# Space radiation damage rescued by inhibition of key spaceflight associated miRNAs

J. Tyson McDonald[1,32], JangKeun Kim[2,32], Lily Farmerie[3,4,32], Meghan L. Johnson[3,32], Nidia S. Trovao[5], Shehbeel Arif[6,7], Keith Siew[8], Sergey Tsoy[9], Yaron Bram[9], Jiwoon Park[2], Eliah Overbey[2], Krista Ryon[2], Jeffrey Haltom[10], Urminder Singh[11], Francisco J. Enguita[12], Victoria Zaksas[13,14], Joseph W. Guarnieri[10], Michael Topper[15], Douglas C. Wallace[10,16], Cem Meydan[2], Stephen Baylin[15], Robert Meller[17], Masafumi Muratani[18,19], D. Marshall Porterfield[20], Brett Kaufman[3,4], Marcelo A. Mori[21,22], Stephen B. Walsh[8], Dominique Sigaudo-Roussel[23], Saida Mebarek[24], Massimo Bottini[25], Christophe A. Marquette[26], Eve Syrkin Wurtele[11,27], Robert E. Schwartz[9], Diego Galeano[28], Christopher E. Mason[2], Peter Grabham[29,33] ✉ & Afshin Beheshti[30,31,33] ✉

Our previous research revealed a key microRNA signature that is associated with spaceflight that can be used as a biomarker and to develop countermeasure treatments to mitigate the damage caused by space radiation. Here, we expand on this work to determine the biological factors rescued by the countermeasure treatment. We performed RNA-sequencing and transcriptomic analysis on 3D microvessel cell cultures exposed to simulated deep space radiation (0.5 Gy of Galactic Cosmic Radiation) with and without the antagonists to three microRNAs: miR-16-5p, miR-125b-5p, and let-7a-5p (*i.e.*, antagomirs). Significant reduction of inflammation and DNA double strand breaks (DSBs) activity and rescue of mitochondria functions are observed after antagomir treatment. Using data from astronaut participants in the NASA Twin Study, Inspiration4, and JAXA missions, we reveal the genes and pathways implicated in the action of these antagomirs are altered in humans. Our findings indicate a countermeasure strategy that can potentially be utilized by astronauts in spaceflight missions to mitigate space radiation damage.

The perils of human space exploration span the gamut of exposure to space radiation, altered gravity fields, and prolonged isolation in a hostile/closed environment[1]. The profound effects on human health may include bone and muscle atrophy, increased intracranial pressure, enhanced risk of developing cancer and cardiovascular disease, and impaired neurocognitive functions[2]. These physical and psychological stressors present a major concern for maintaining the overall health of spaceflight crews. The biological consequences of spaceflight also present a unique opportunity to analyze aspects of natural and disease pathologies that occur terrestrially[3–5].

Circulating microRNAs (miRNAs) critically impact cell non-autonomous regulation of post-transcriptional processes, can be used as biomarkers[6,7], and are associated with human diseases[8–10]. We have extensively characterized a circulating miRNA signature associated with the space environment using biomedical samples from astronauts, in vivo rodent models, in vitro cellular models, and 3D

human-derived cell cultures under simulated and real space exposure[11–14]. Transcriptome analyses demonstrated expression of candidate miRNAs were altered in liver, kidney, adrenal gland, thymus, mammary gland, skin, and skeletal muscle from space-flown animal models[11] and revealed a conserved miRNA signature between rodents and humans[13]. The altered miRNA signature discovered after exposure to space irradiation was predicted to affect the cardiovascular system and was widely associated with mitochondrial dysfunction[13]. Treatment of 3D microvessel cell cultures with miRNA inhibitors (referred to as antagomirs) targeting three of these miRNAs, miR-16-5p, miR-125b-5p, and let-7a-5p, prevented microvessel collapse[13] and rescued the inhibition of angiogenesis[15] caused by space radiation exposure.

Multi-omic analyses of biomedical profiles of astronauts along with several hundred space-related data points showed that major shifts in mitochondrial gene expression may be central to spaceflight-associated health risks[12]. Mitochondria are responsible for metabolic activity through oxidative phosphorylation (OXPHOS), and mitochondrial dysfunction decreases energy availability, alters cellular metabolism, and leads to cancer, diabetes, heart disease, and muscular and neurological disorders[16].

Here, a 3D model for human microvessels is used to explore the cellular biology consequences of miRNA inhibition, particularly focusing on how the endothelial cell layer may be involved with cardiovascular damage. We find that antagomir treatment reduces DNA damage induced by space radiation, and is associated with improved stress responses at the molecular and morphological levels. Furthermore, we confirm the specific targets of the miRNAs in astronaut samples from the NASA Twin Study[17], a JAXA mission[18], and the first civilian commercial spaceflight mission, Inspiration4 (I4)[19]. Our results suggest that these miRNAs may be targeted as countermeasures to alter cellular signaling pathways to mitigate spaceflight damage and reduce human health risks in human spaceflight missions.

## Results

### Combined antagomir inhibition is effective in counteracting space radiation damage

We investigated the cellular alterations associated with the protective roles of three antagomirs targeting miR-16-5p, miR-125b-5p, and let7a-5p in a 3D human microvasculature model. Firstly, we examined the effect of individual miRNAs to mitigate structural damage caused by space radiation exposure from simulated Galactic Cosmic Rays (GCR). The morphology of mature microvessels representing post-mitotic cells with characteristics of the endothelial barrier (tight junctions) was preserved by inhibiting any of the three key miRNAs (Fig. 1a, c). A single dose of each individual antagomir 24 h before GCR irradiation was sufficient to protect microvessels although the combination of the three was more effective. Furthermore, the inhibition by each miRNA was comparable to the unirradiated controls.

A similar result was found for the longer multistage development of microvessels undergoing angiogenesis. In this model, cells were exposed to GCR on culture day 1, when they were still individual and not connected; while the irradiation effects were seen on day 6, when the endothelial cells should have been formed into mature microvessels (Fig. 1b, c). Individual antagomirs applied on days 2 and 3 restore angiogenesis at levels comparable to controls, and combined antagomirs exert an improved protective role (Fig. 1b). In both models, miR-16-5p, miR-125b-5p, and let-7a-5p are similarly effective at counteracting the effects of space radiation, but the antagomir combination inhibition of the three miRNAs provides a stronger and more robust response. Hereafter, we focus on the biological changes occurring when inhibiting all three miRNAs.

### Combined antagomir inhibition protects microvessels from DNA double-strand breaks (DSBs) induced by space radiation

While the biological consequences of GCR exposure on mature and developing microvessels are clearly observed 24 h and 7 days after irradiation, respectively, examination of earlier endpoints would enable us to understand the impact of antagomir treatment. Since DNA is the primary target damaged by radiation, early effects from miRNA inhibition on DNA damage and repair were investigated.

As the repair of DNA DSB damage by space radiation is pivotal for cellular function, we assessed the cellular proteins involved in DNA DSB repair targeted by miRNA inhibition. Many of the DNA DSB repair genes are common targets of the miRNAs as shown by a network representation for the perturbation effects of miR-125b-5p, miR-16-5p, and let-7a-5p (Fig. 2a). These results suggest that miRNAs play an important role in activating the DNA DSB repair pathways induced by GCR irradiation.

To determine the molecular mechanism through which the antagomirs protect mature and developing microvessels from the effects of GCR, we studied the accumulation of DNA DSBs and the activation of DNA repair pathways. Immunocytochemical staining for the tumor suppressor p53 binding protein 1 (53BP1) was quantified following antagomir treatment 90 min after GCR irradiation (Fig. 2b). The 53BP1 protein accumulates at the site DNA DSBs after irradiation, and the resulting repair foci can be measured as a critical platform for DNA DSB repair activity[20]. The peak DNA repair foci often occurs around 30 min after gamma irradiation and our previous publications showed that foci induced by charged particles decline over several hours in the human microvasculature model[21]. Our current analysis shows a significant increase in DNA DSB repair foci (Fig. 2b). In contrast, the antagomir treatment significantly decreased the number of foci to levels similar to the unirradiated condition. This demonstrates that the antagomirs protect microvasculature cells from DNA DSBs caused by space radiation at much earlier times due to potentially providing a more efficient DNA repair mechanism.

The changes in expression of miRNAs after GCR exposure and its relationship to DNA repair mechanisms is not widely understood[22]. To determine if DNA DSBs are associated with a global response from those evolutionarily conserved miRNAs that we observed in vivo, we exposed C57BL/6 female mice to 0.5 Gy of GCR and performed miRNA-sequencing on the plasma, liver, soleus muscle, and heart (Fig. 2c, d). From pathway analysis[23] of DNA DSB and repair pathways, we observed a comprehensive increase in DNA DSB genesets across all four tissues from the measured changes in miRNA expression. The global increase associated with overall miRNA activity following GCR irradiation further supports the key roles of these miRNAs in the DNA DSB repair pathways.

### Space radiation reveals key regulation of the three miRNAs in specific tissues

To further establish miRNA changes after space radiation and the importance of miR-16, miR-125b, and let-7a with cardiovascular health risks, we performed weighted gene co-expression network analysis (WGCNA) on the miRNA-seq data from the mice exposed to 0.5 Gy of GCR (Fig. 3)[24]. WGCNA classically was designed for transcriptomic RNA-seq data, but recently has proven its utility in miRNA-seq data[25,26]. Hence, we utilized WGCNA to determine key miRNA networks that are impacted by space radiation and how they are affected in the heart, liver, soleus muscle, and plasma of mice. The WGCNA analysis revealed three distinct miRNA networks (i.e., "Modules" or "ME") that are most strongly impacted by GCR radiation, depending on the tissue type (Supplementary Fig. 1 and Fig. 3a–c). Interestingly, for all three Modules the heart tissue showed the most significant decrease comparing GCR irradiated mice to sham (Fig. 3a–c). For Module 0 (Figs. 3a) and 2 (Fig. 3c), the liver shows a significant increase. When observing the key

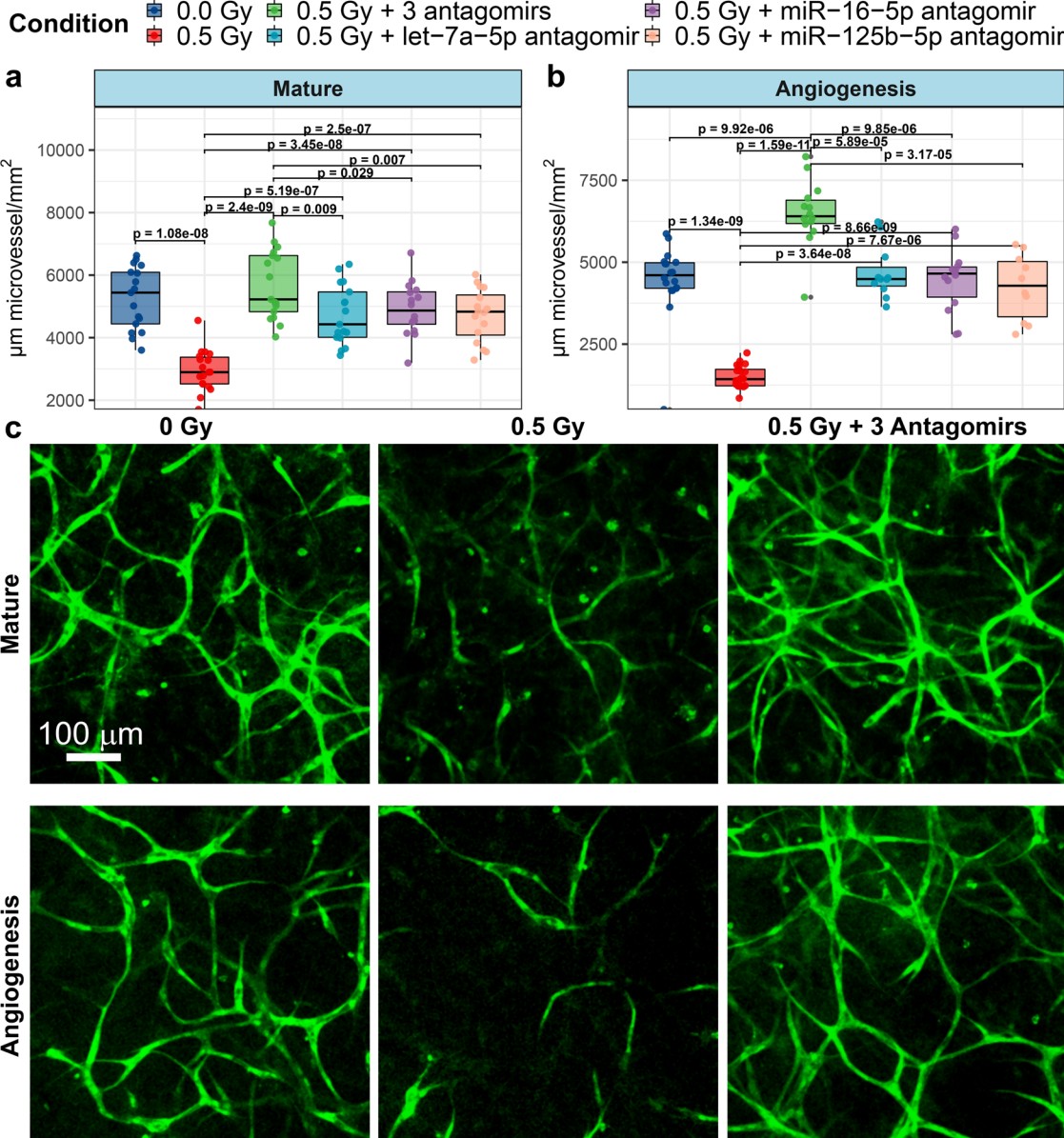

**Fig. 1 | Antagomir treatment of 3D human umbilical vein endothelial cell (HUVEC) microvessel cell cultures. a, b** Quantification of formation of microvessels using 3D cultures of human microvessels irradiated with 0.5 Gy of simplified simulated galactic cosmic rays (GCR) compared to sham irradiated samples. Irradiated cultures were treated with or without antagomir-induced inhibition of miR-125b-5p, miR-16-5p, and/or let-7a-5p. For the boxplots the center line represents the median and the lines extending from both ends of the box indicates the quartile (Q) variability outside Q1 and Q3 to the minimum and maximum values. **a** mature and **b** angiogenesis microvessels. The p-values were determined by two-side multiple pairwise comparison. **c** Mature and angiogenesis microvessels fixed and fluorescently stained with 5-(4,6-dichlorotriazinyl) aminofluorescein (DTAF) after GCR irradiation, with or without antagomirs 24 h prior to irradiation. Scale bar = 100 μm. For mature the following biological independent samples were used: $n = 17$ for 0 Gy, $n = 18$ for 0.5 Gy, $n = 17$ for 0.5 Gy + 3 antagomirs, $n = 17$ for 0.5 Gy + let-7a-5p antagomir, $n = 17$ for 0.5 Gy + miR-16-5p antagomir, and $n = 16$ for 0.5 Gy + miR-125b-5p antagomir. For angiogenesis the following biological independent samples were performed: $n = 18$ for 0 Gy, $n = 20$ for 0.5 Gy, $n = 14$ for 0.5 Gy + 3 antagomirs, $n = 11$ for 0.5 Gy + let-7a-5p antagomir, $n = 14$ for 0.5 Gy + miR-16-5p antagomir, and $n = 10$ for 0.5 Gy + miR-125b-5p antagomir.

miRNAs involved in each module we see that let-7a-5p is present in Module 0 (Fig. 3d), miR-16-5p in Module 1 (Fig. 3e), and miR-15b-5p in Module 2 (Fig. 3f). The individual miRNAs within each Module were differential expressed due to GCR radiation (Supplementary Fig. 2).

**Rescue of space radiation damage with inhibition of key miRNAs associated with cardiovascular space radiation damage**

Based on our intriguing results showing complete protection of the 3D human microvessel cell cultures post GCR irradiation by the inhibition

of three miRNAs, we further investigated the key biological processes involved. We performed bulk RNA-sequencing (RNA-seq) on the mature microvessel and angiogenesis models after 0.5 Gy of GCR exposure and compared conditions with or without the combination of antagomirs (Fig. 4). The experimental design for the antagomir treatment is depicted in Fig. 4a. The antagomir treatment and endpoints after irradiation differ between the two models as described in Malkani et al.[13] for the mature microvessel model and Wuu et al.[15] for the angiogenesis model.

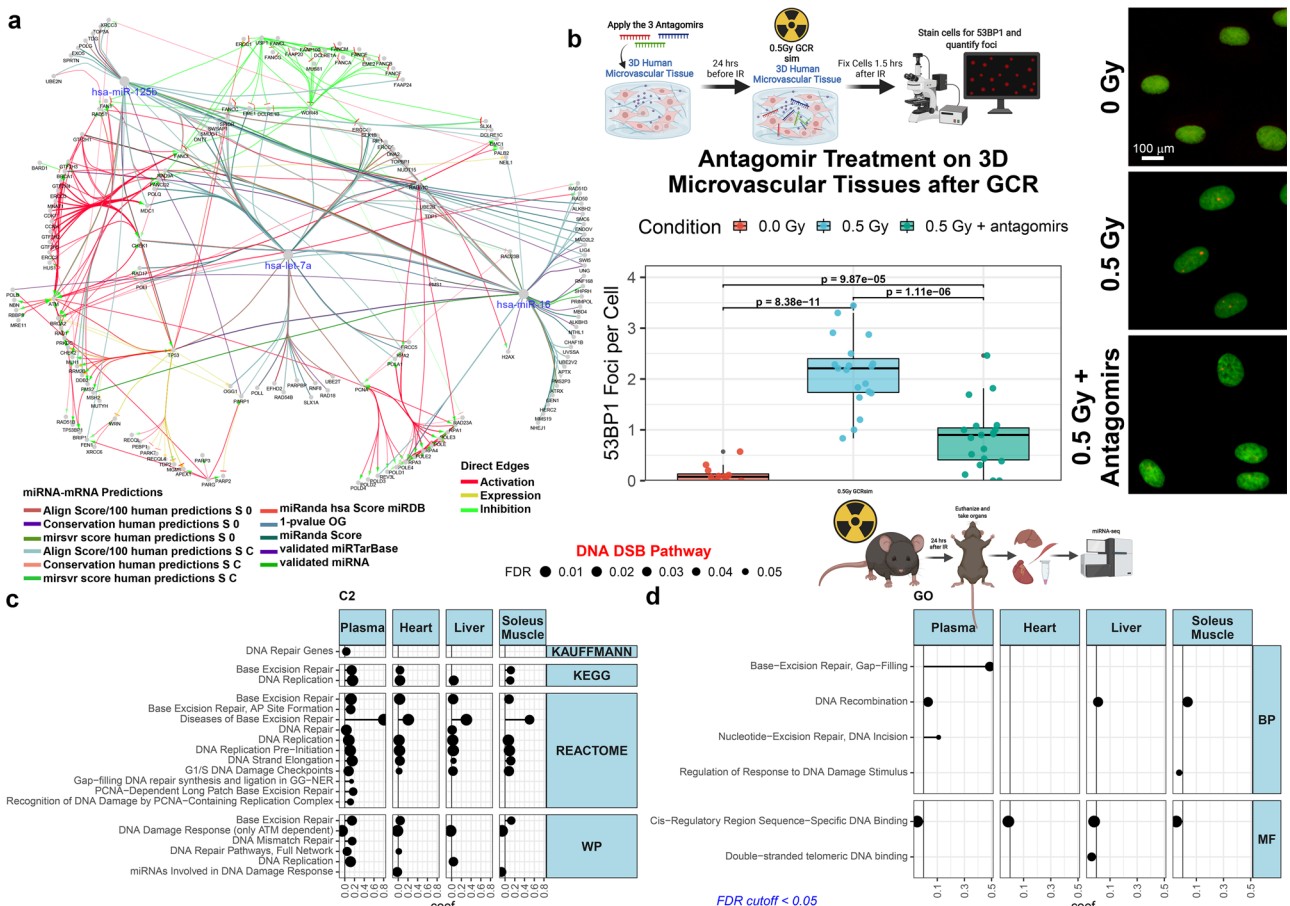

**Fig. 2 | Quantification of DNA double strand break (DSB) repair foci with antagomir treatment and miRNA pathway analysis. a** Network representation for the DNA DSB repair gene targets for miR-125b-5p, miR-16-5p, and let-7a-5p generated by ClueGO in Cytoscape. As indicated in the figure legend, the color for the edges indicate either the predictions used for the miRNA-mRNA connection or the influence two different nodes will have on each other (i.e. Direct Edges).
**b** Quantification of 53BP1 DNA repair foci in the mature 3D HUVEC microvessel cell culture 1.5 h after irradiation with 0.5 Gy of GCR. Representative images are shown on the right (scale bar = 20 μm). The p-values were determined by two-side multiple pairwise comparison. *n* = 3 biologically independent samples examined for each conditions and a total of the following random independent field of views for each condition: *n* = 12 field of views for 0 Gy and *n* = 19 field of views for both 0.5 Gy and

0.5 Gy + antagomirs. For the boxplot the center line represents the median and the lines extending from both ends of the box indicates the quartile (Q) variability outside Q1 and Q3 to the minimum and maximum values. The schematic of the experiment was created with BioRender.com. DNA DSB pathway-specific Gene Set Enrichment Analysis (GSEA) from **c** the curated chemical and genetic perturbations and canonical pathways collection (C2) and **d** the gene ontology (GO) collection using miRNA-sequencing data from different tissues (i.e. liver, heart, soleus muscle, and plasma) from C57BL/6 female mice irradiated with or without 0.5 Gy OF GCR exposure. Mice were euthanized (*N* = 10 irradiated and *N* = 10 sham controls) and tissues were harvested 24 h after irradiation. In Fig. 2 the schematics in panels **b** and **d** created with BioRender.com released under a Creative Commons Attribution-NonCommercial-NoDerivs 4.0 International license.

Global transcriptomic analysis of RNA-seq results revealed key patterns associated to antagomir treatment after GCR irradiation. Overall, antagomir treated samples post-irradiation clustered closer to the sham unirradiated samples for both microvessel cell cultures (Fig. 4b). This indicates that the antagomir treatment had a substantial impact on restoring gene expression back to normal levels. Of note, we are utilizing *p*-value statistics for our main analysis and acknowledge that this may result in the potential of more false positives, but we believe that the information generated by lowering the statistical significance will produce more meaningful data and previous literature also provided further justification when necessary to utilize *p*-values[27,28]. In addition, we have also provided the adj. *p*-value analysis for full transparency of our results (Supplementary Fig. 3).

Although the global clustering of both models demonstrates a restoration closer to the basal levels, there are measurable differences between the two models with GCR irradiation alone and GCR with the antagomirs. This is apparent when observing the transcriptomic patterns for the significantly regulated genes (Fig. 4c). GCR irradiation caused a greater number of genes to be differentially regulated for the

mature vessel compared to the angiogenesis model (Fig. 4d, e). Antagomir treatment during irradiation in mature (before) or in angiogenesis cells (after) caused significant dysregulation in both models (Fig. 4d, e). Substantially more genes were uniquely altered in the angiogenesis compared to the mature microvessel model (Fig. 4f), as may be expected since the endpoint is much longer than in the mature microvessel model and encompasses multiple stages of development.

We also identified a set of orphan genes to be differentially regulated in GCR-stressed samples and reversed by antagomir treatment. Orphan genes code for proteins that are unique to a species[29–31] and may provide a major source of evolutionary novelty conferring beneficial traits to a species. Evidence-based (EB) orphan genes were identified by Singh[32] based on a comprehensive analysis of 27,000 RNA-Seq datasets from the Cancer Genome Atlas and Genotype-Tissue Expression project. To determine how human orphan genes respond to space radiation and antagomir treatment, differential expression of orphan genes was assessed in the RNA-seq data. In both models, tens of orphan genes were differentially

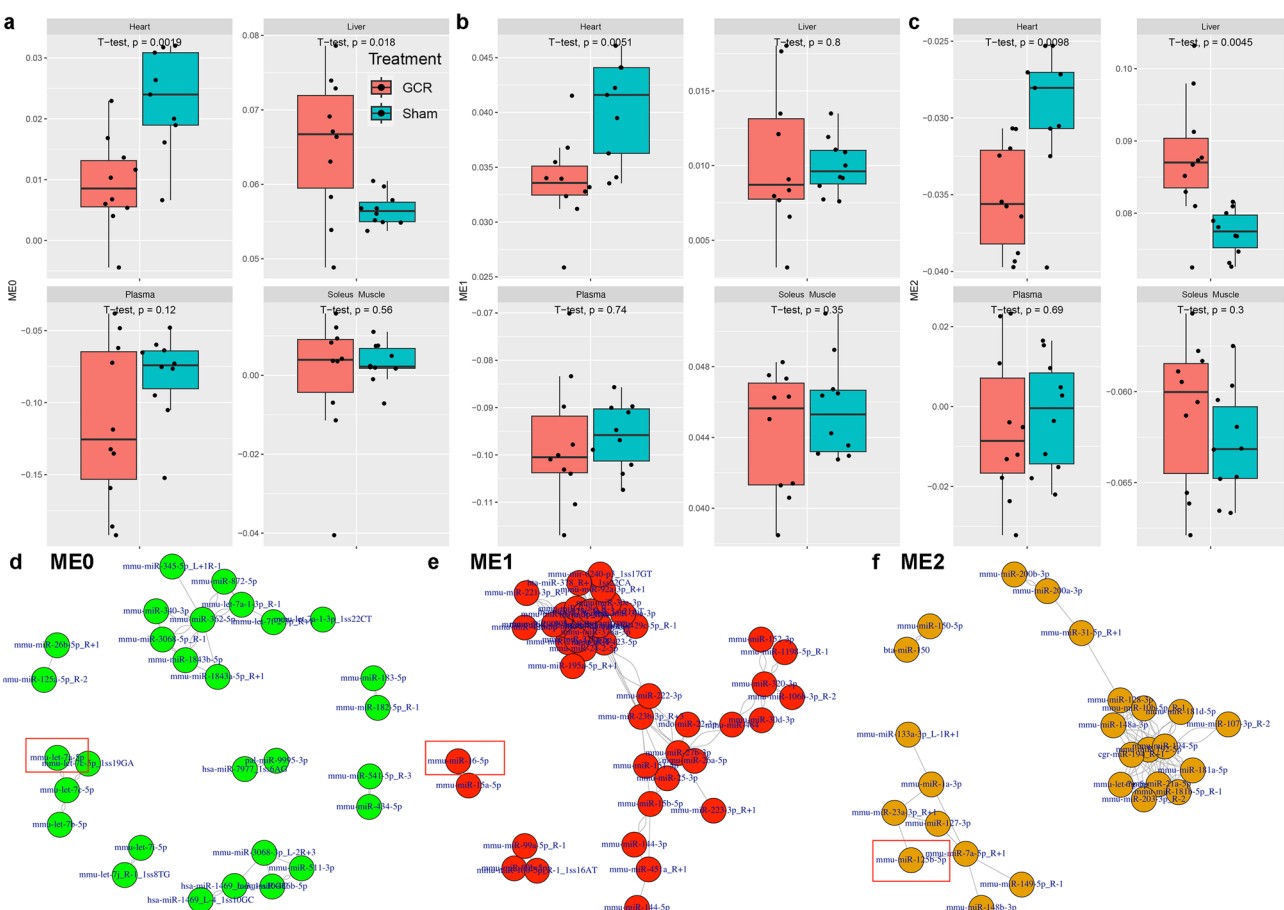

**Fig. 3 | MiRNA based weighted gene co-expression network analysis (WGCNA) on different organs from GCR irradiated mice.** Boxplots displaying the expression of co-expressed miRNA network **a** Module 0 (ME0), **b** Module 1 (ME1), and **c** Module 2 (ME2) in heart, liver, plasma, and soleus muscle samples of mice that underwent Sham or GCR treatment. The y-axis represents the eigengene expression values of each network module. For the boxplots the center line represents the median and the lines extending from both ends of the box indicates the quartile (Q) variability outside Q1 and Q3 to the minimum and maximum values. The p-values were determined by two-side pairwise comparison. Network representation of highly correlated co-expressed miRNAs for **d** ME0, **e** ME1, and **f** ME2. The red box annotations highlight let-7a-5p, miR-16-5p, and miR-125b-5p. For all murine experiments $n = 10$ biologically independent animals examined with one beam time.

expressed by space irradiation plus antagomirs, but not with radiation alone (Fig. 4c, d, and Supplementary Fig. 4). Since orphan genes often integrate themselves into metabolic and regulatory networks[31,33,34], these could play a role in the dysregulation of DNA repair and metabolic pathways associated with response to space radiation and antagomir treatment observed here.

To better identify dysregulated genes targeted by the antagomirs, we used miRNA target analysis by the mirDIP algorithm[35] to assess transcripts downregulated by irradiation but restored after antagomir treatment. Twenty-one gene targets, including *PDPR*, *APAF1*, *XPO6*, *NIBAN1*, and *SLC7A1*, were predicted to be simultaneous targets of each miRNA (Fig. 5a). To validate the key genes, we performed PCR on 8 of the 21 genes from the same isolated RNA utilized for the RNA-seq (Fig. 5b). We see that all the genes are in agreement with the RNA-seq data except for *MSH5*, which we see an upregulation with 0.5 Gy and then the expression returns back to 0 Gy levels after antagomir treatment. We investigated how well the miRNAs bound to the 21 gene targets ("antagomir-rescued genes"), and the log$_2$ fold change from the RNA-seq data after GCR exposure. We observed that the three miRNAs bound to the 3′-UTR and CDS regions of 15 (Fig. 5c) and 13 (Fig. 5d) out of the 21 antagomir-rescued genes, respectively, whereas let-7a-5p and miR-125b-5p bound only to two of the target genes (Fig. 5e).

To survey the efficacy of the three miRNAs to downregulate the 21 gene targets, we analyzed RNA-seq fold change values from different experimental conditions using cumulative plots[36] see Methods). Figure 5f shows that the 21 antagomir-rescued genes with 7-8mer 3′-UTR sites had a significant propensity to be downregulated compared to those without 3′-UTR sites ($P = 4.99e-07$, $2.65e-32$ and $6.31e-09$ for 8mer, 7mer-m8 and 7mer-A1, respectively, using the Kolmogorov–Smirnov test). These results indicate that miR-16-5p, miR-125b-5p, and let7a-5p play a key role in the downregulation of the 21 antagomir-rescued genes. Figure 5g, h shows that the downregulation of these genes is drastically reduced under antagomir treatment, indicating that by inhibiting these 3 miRNAs, expression of these genes reverts towards the control (0 Gy) levels.

Pathway analysis of the antagomir-restored transcripts targeted by one or more of the three miRNA shows an enrichment pattern for expression of genes associated with apoptosis and TP53-mediated cell death, cell cycle and mitotic phase transitions, innate immune response, and mitochondrial processes related with amino acids (Supplementary Fig. 5). Thus, alteration of these pathways may account for the observed preservation of the 3D microvessels by antagomir treatment with GCR exposure.

To confirm that these 21 genes are actual targets for the three miRNAs we dissected the pathways involved with the mirDIP algorithm analysis and also provided experimental evidence from current databases and literature. While mirDIP draws from a comprehensive pool of 24 databases to evaluate miRNA targets[35],

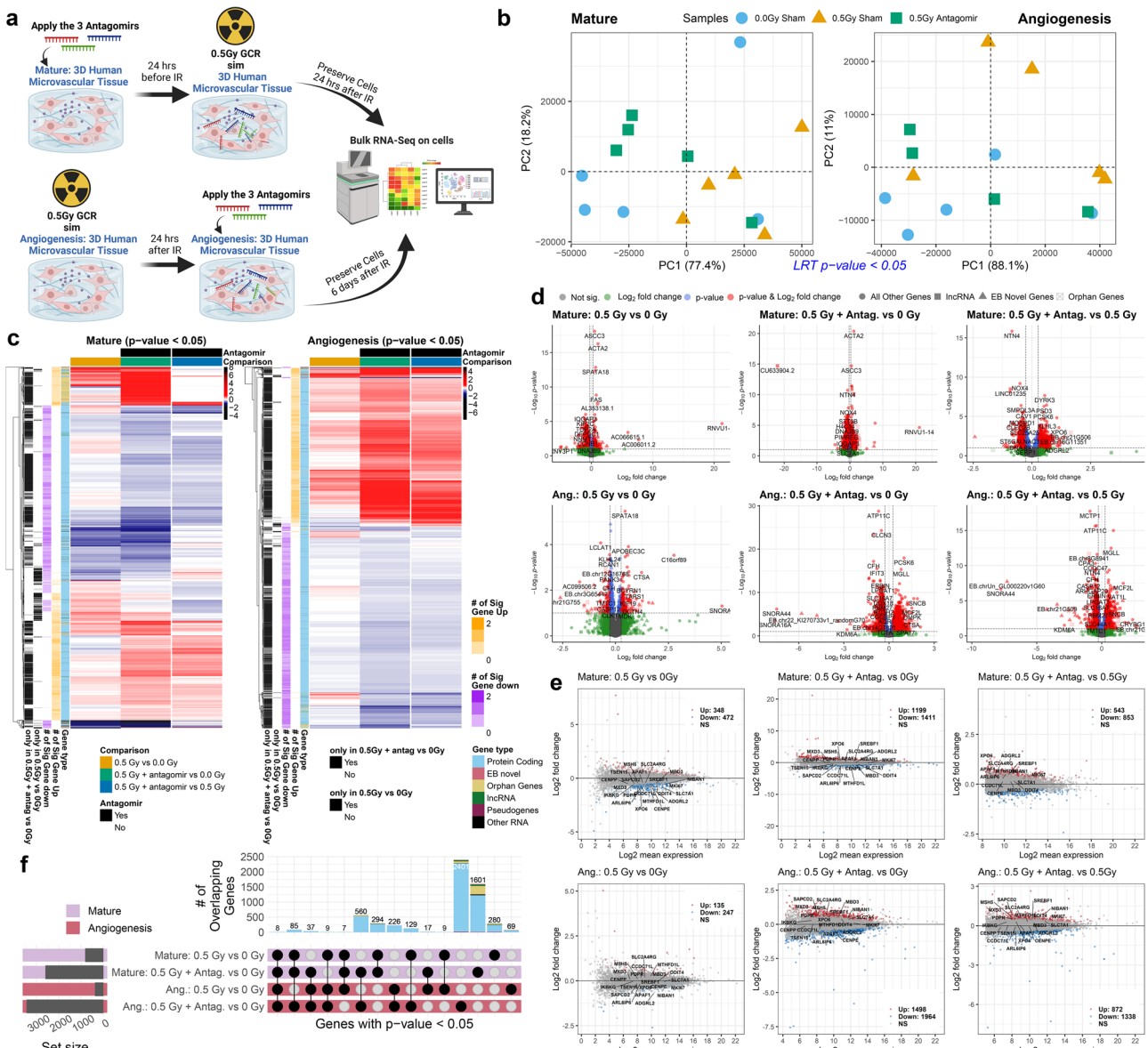

**Fig. 4 | Global transcriptomic analysis reveals that antagomirs revert gene profiles closer to control samples. a** Schematic of the experimental profile for the mature and angiogenesis 3D cell culture models irradiated with 0.5 Gy of GCR and prepared for RNA-sequencing. There was $n = 5$ biological replicates for each condition. Created with BioRender.com released under a Creative Commons Attribution-NonCommercial-NoDerivs 4.0 International license. **b** Principal Component Analysis (PCA) of the significantly regulated genes ($p$-value < 0.05) for all conditions compared with likelihood ratio test (LTR) analysis for both mature and angiogenesis 3D microvessel cell culture models. **c** Heatmap of significantly regulated genes ($p$-value < 0.05) for mature and angiogenesis cell culture models. For each gene (i.e. row), the criteria for display was to have at least one comparison per gene to be significantly regulated. Then the trends for non-significant genes for that row were also displayed. The $\log_2$(fold-change) values are displayed. The side color bars indicate the number of significant genes that are either up- or down-regulated per row and also the type of gene. **d** Volcano and **e** MA plots for each comparison. **f** Upset plot displaying the overlapping significantly regulated genes ($p$-value < 0.05) for mature and angiogenesis models with and without antagomir treatment. For all RNA-seq data Wald test and the likelihood ratio test was used to generate the F statistic $p$-value. The adj. p-value plots are also provided in Supplementary Fig. 3.

our focus was on the analysis of the six most reputable databases among them (Fig. 5i). Additionally, we corroborated the experimental validation of these gene targets by cross-referencing data from three databases (i.e. miRTarBase[37], TarBase[38], and starBase[39]) alongside pertinent literature references (Fig. 5i and Supplementary Table 1). Our results indicate robust experimental validation of gene targets for the three miRNAs as follows: 10 out of 10 for miR-16-5p, 13 out of 14 for let-7a-5p, and 9 out of 15 for miR-125b-5p. It's worth noting that while the validation rate for miR-125b-5p stands at 60% based on these databases, certain contextual conditions may influence the validation of gene targets for miRNAs. Nevertheless,

our additional experimental evidence (Fig. 5b–h) enhances the confidence in our findings.

Finally, it is crucial to assess the conservation of these three miRNAs between humans and mice. It is well-established that miRNAs exhibit high levels of conservation across species, particularly between mice and humans[40]. Our analysis demonstrates 100% conservation of all three mature miRNAs (i.e. miR-16-5p, miR-125p-5p, and let-7a-5p) and their precursor forms have conservation rates greater than 95% between mouse and human genomes (Fig. 5j). Given the robust conservation observed, comparisons between results obtained in mouse and human contexts can be easily facilitated.

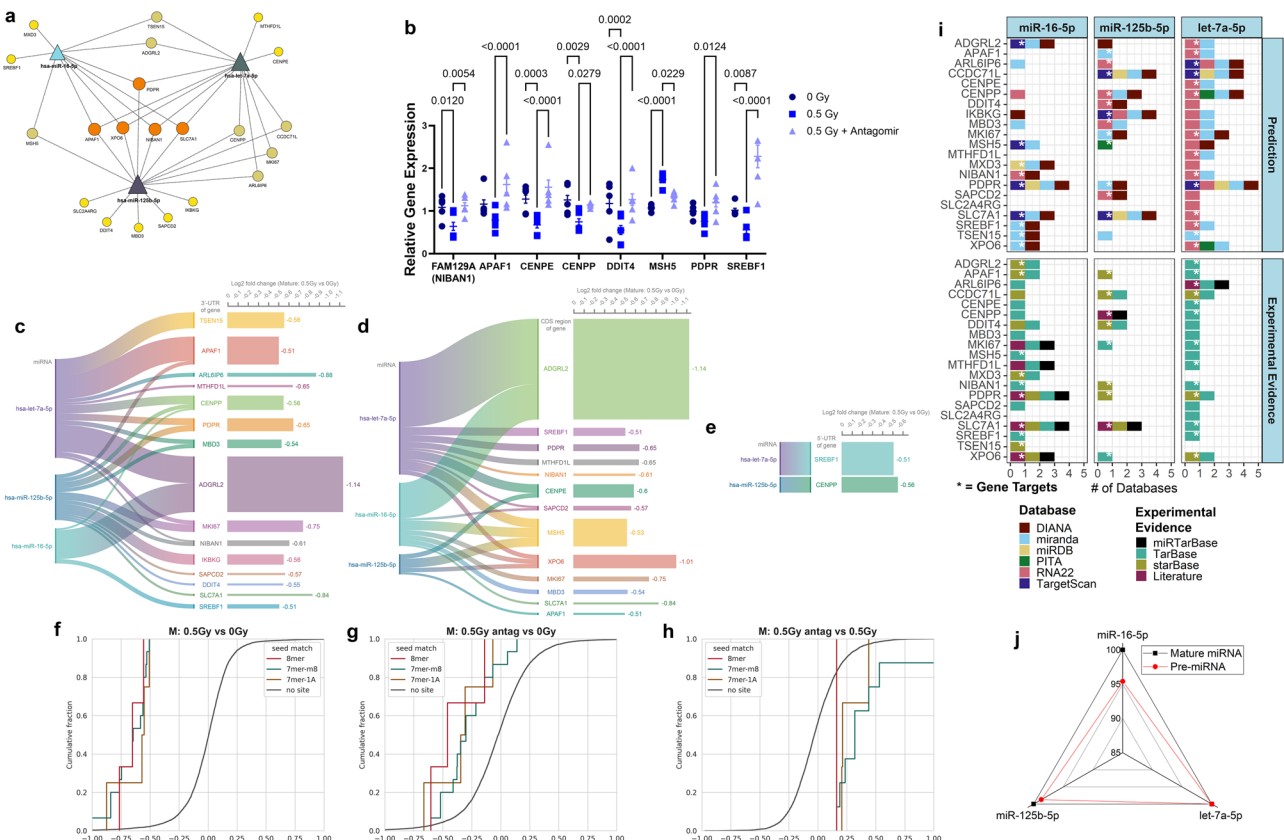

**Fig. 5 | 21 key gene targets for the three miRNAs restored back to control levels after antagomir treatment. a** Network display of the predicted miRNA gene targets by the mirDIP algorithm that were significantly regulated (*p*-value < 0.05) for 0.5 Gy versus 0 Gy, but were not statistically significant for 0.5 Gy with the antagomirs versus 0 Gy in the mature 3D cell culture model. For all RNA-seq data Wald test and the likelihood ratio test was used to generate the F statistic *p*-value. **b** Relative Gene Expression from PCR validation of 8 out of the 21 key genes. The *p*-values were determined by two-side multiple pairwise comparison. Data are presented as a dot plot of individual values with the mean values indicated with the +/−SEM. Binding probability of miRNAs to **c** 3′-UTR, **d** CDS, and **e** 5′-UTR of gene targets and their log$_2$(fold change) in the presence or absence of radiation. Cumulative plots for the 21 key genes with comparing 0.5 Gy vs 0 Gy (**f**), 0.5 Gy +

antagomir vs 0 Gy (**g**), and 0.5 Gy + antagomir vs 0.5 Gy (**h**) for the 3D mature microvessels. **i** Bar plot of the databases utilized by mirDIP (i.e. DIANA, miranda, miRDB, PITA, RNA22, and TargetScan) and experimental evidence for the 21 genes targeted by miR-16-5p, miR-125b-5p, and let-7a-5p. The experimental evidence for the targets have been compiled from TarBase, miRTarBase, starBase, and literature. We have also provided the details for the experimental evidence from these databases in Supplementary Table 1. The *mirDIP gene targets for each miRNA as shown in panel **a**. It is important to note that the databases provide literature references for all the experimental evidence and the literature indication in the figure is for additional literature evidence outside of these databases. **j** Radar plot illustrating homology of pre-miRNAs (red) and mature miRNAs (black) in humans and mice for miR-16-5p, miR-125b-5p, and let-7a-5p.

## Regulation of antagomir-rescued genes in astronauts

The relevance to humans of the 21 genes that are rescued by the antagomir countermeasures in the 3D human tissues (Fig. 6a) were further examined using data from the NASA Twin Study[17] (REF) (Fig. 6a–d), Japanese Space Agency (JAXA) Cell-Free Epigenome (CFE) Study[18] of astronauts that were in space for 120 days (Fig. 6e, f), as well as data from the first civilian commercial 3-day spaceflight mission, referred to as Inspiration4 (I4)[19] (Fig. 6g–s). The NASA Twin Study has been described in several papers which compared two identical twins with one Twin on the ISS for 340 days and the other twin on Earth[12,17]. The JAXA study conducted RNA-seq on plasma cell-free RNAs from six astronauts who were on the International Space Station (ISS) for 120 days, sampled from pre-flight, during flight, and post-flight time points. The I4 mission performed single-cell RNA-seq (scRNA-seq) on the blood and also RNA-seq skin biopsies which involved 4 astronauts (two males and two females) that orbited at low Earth orbit (LEO) at 590 km above the atmosphere (Fig. 6f). The I4 mission occurred at a much higher altitude than the ISS, thus the I4 astronauts experienced a higher accumulated dose of radiation compared to being on the ISS. It is important to note that the estimated accumulated doses of space radiation that the astronauts were exposed to for these missions were

lower than 0.5 Gy (which is the estimated dose for a trip to Mars and back), but despite this we still observed key changes as we will describe below. The following are the accumulated doses that were measured for each mission: NASA Twin Study astronaut received 146.34 mSv or 14.634 cGy[17], I4 astronauts received 4.72 mSv or 0.472 cGy[19], and for the JAXA astronauts we estimated that they had received an accumulated dose of 51.60 mSv or 5.16 cGy (the dosimetry was not reported for this mission and this is an estimate based on the average dose received per month from other missions)[18].

Using RNA-seq data from the NASA Twin Study, we analyzed the expression of the 21 antagomir-rescued genes over four different circulating cell populations with the Twin onboard the ISS for 340 days: CD4 T cells (Fig. 6a), CD8 T cells (Fig. 6b), CD19 B cells (Fig. 6c), and a lymphocyte depleted cell population which will represent the monocytes (Fig. 6d). There are individual gene variations observed from preflight to the last time point after return (i.e. R + 200 days), but the overall gene signature changes will indicate how these genes will be impacted by the three miRNAs. The global changes for these 21 genes over time is best observed with cumulative plots for the different cell types for the Twin in space over time (Supplementary Fig. 7a and Fig. 7a). It was observed that the only cell type to show significant

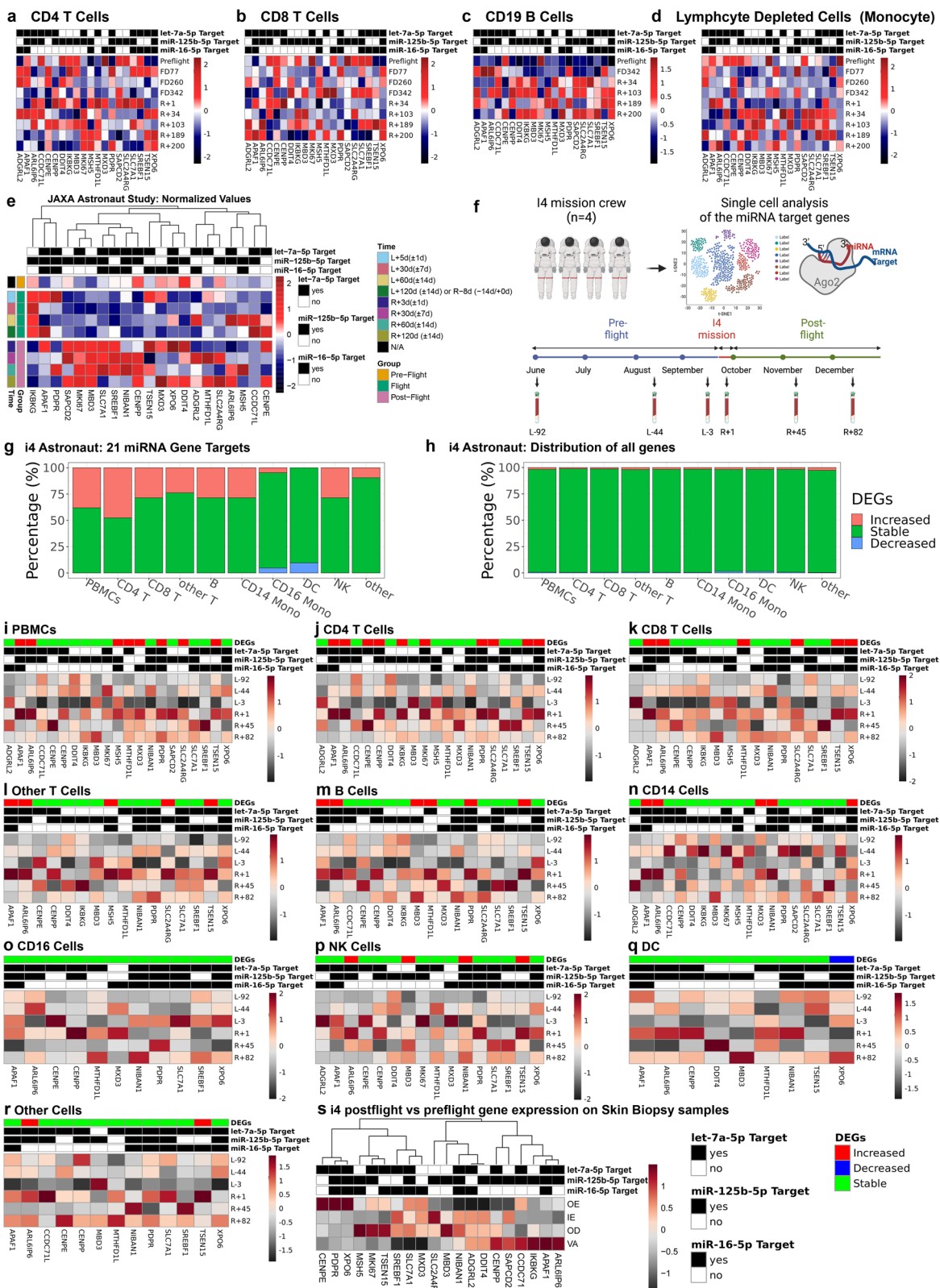

changes for the 21 genes from flight to return on Earth were the lymphocyte depleted cells or monocytes (Fig. 7a). We show that for flight vs pre-flight and R + 1 vs pre-flight (i.e. immediately after return to Earth) there was significant downregulation for these key genes and then at 34 days and beyond, the genes have returned back to the control pre-flight expression.

To establish the direct relationship between the 21 gene targets and the three miRNAs, we utilized the NASA Twin Study which had both miRNA- and mRNA-seq data associated with the mission and cell types. Similar to the cumulative plots, we observed the correlation between the three miRNAs (i.e. miR-16, let-7a, and miR-125b) with the 21 gene targets in the four different cell types (i.e. CD4, CD8, CD19, and

**Fig. 6 | Expression of 21 genes rescued by the antagomir treatment in astronauts from the NASA Twin Study, JAXA CFE, and Inspiration4 (i4) missions. a–d** Heatmaps displaying the 21 key gene profile for the different cell types (*i.e.* CD4 T cells, CD8 T cells, CD19 B cells, and lymphocyte deleted cell population) from RNA-seq on the NASA Twin Study. **e** Heatmap of the normalized plasma cell-free RNA expression values for the 21 key genes over time for the six astronauts over 120 days in space from JAXA study. The values shown are the averaged normalized expression values for all six astronauts for each time point during flight and post-flight. The three pre-flight time points were averaged together, since the changes for genes in the time leading up to flight are considered to be the same and part of the baseline values. For the time, L = Launch (i.e. meaning time after launch from

Earth and length in space) and R = Return to Earth. **f** A schematic of the i4 experimental design created with BioRender.com released under a Creative Commons Attribution-NonCommercial-NoDerivs 4.0 International license. Global gene expression profile of the **g** 21 key genes and **h** all genes from scRNA-sequence data from the i4 astronauts. **i–r** Heatmaps displaying the 21 key gene profile for the different cell types (i.e. PBMCs, CD4 T cells, CD8 T cells, Other T cells, B cells, CD14 cells, CD16 cells, NK cells, Dendritic cells (DC), and other cells) from scRNA-seq on the i4 astronauts. **s** Heatmap visualization of the 21 key genes on the i4 astronauts of relative expression changes in the postflight relative to preflight skin biopsy sample across four skin compartments including Outer Epidermis (OE), Inner Epidermis (IE), Outer Dermis (OD), and Vasculature (VA).

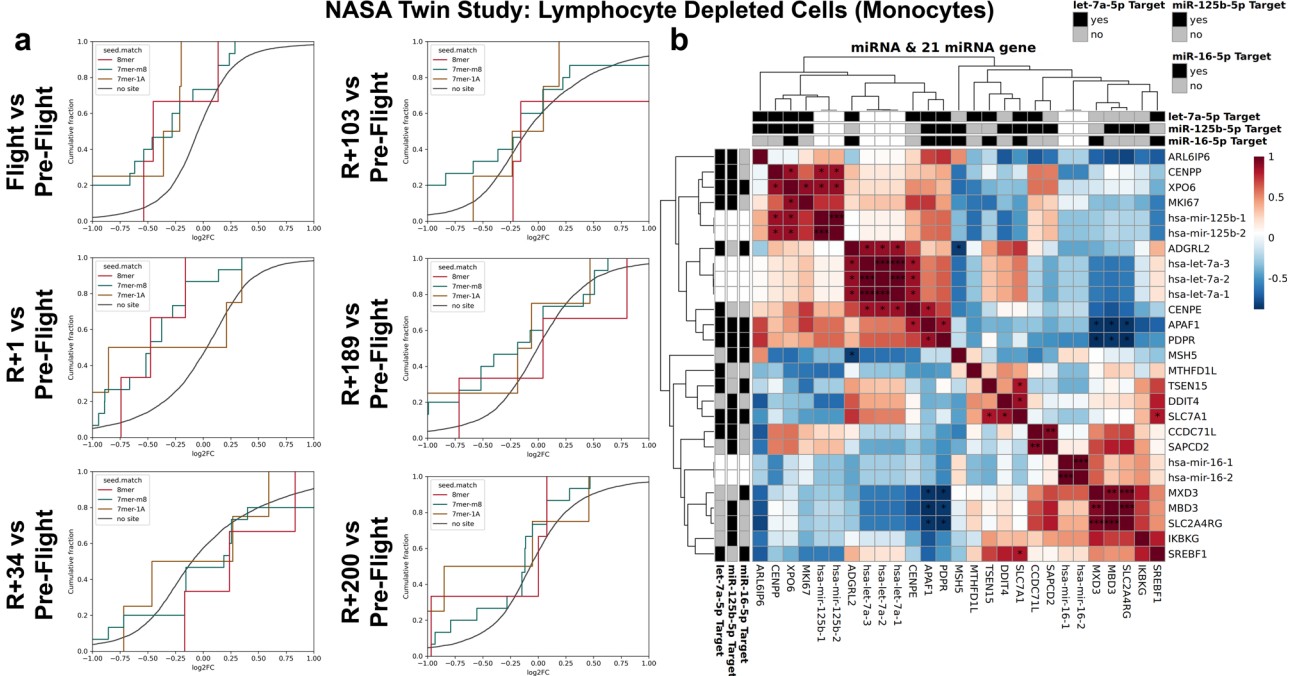

**Fig. 7 | The interaction of the three miRNAs with the 21 gene targets from the NASA Twin Study in Monocytes. a** Cumulative plots for change over time with the 21 genes targeted by the three miRNAs using RNA-seq data from the NASA Twin Study data. Sequencing was performed on lymphocyte depleted cells (i.e. monocytes) from the Twin that was onboard the ISS for 340 days. There are comparisons over time starting with Flight versus Pre-Flight as well as comparing the different Return (R) to Earth timepoints from 1 to 200 days versus Pre-Flight. **b** Correlation

plot of the three miRNAs (i.e. miR-16-5p, let-7a-5p, and miR-125b-5p) and the 21 gene targets for the overlapping time points from the miRNA-seq and mRNA-seq NASA Twin Study data. The gene targets for the miRNAs are indicated in the outer black (yes for gene target) and gray (no for gene target) rows and columns. Wald test and the likelihood ratio test was used to generate the F statistic *p*-value and significance for the correlation is shown by *$p$-value < 0.05.

monocytes) (Supplementary Fig. 7b). We correlated with the overlapping time points per each sample type and observed for the majority of the miRNAs and genes targets an inverse relationship. Specifically, for monocytes for the specific gene targets the miRNAs had an inverse correlation (Fig. 7b). Since the miRNAs will traditionally silence the specific gene target, this result is what we would expect and provides further validation in astronauts that these miRNAs in circulation will be ideal targets to reduce health risks as our in vitro results show.

We noticed that most of the 21 antagomir-rescued genes in the JAXA data were downregulated during the flight, indicating that increased activity of the let-7a-5p, mir-125b-5p, and mir-16-5p might cause the suppression of these target genes in astronauts (Fig. 6e). Most of the genes show up-regulation post-flight as compared to pre-flight and during flight (Fig. 6e). This is consistent with the i4 data, in which antagomir-rescued gene activity is significantly increased post-flight compared to the pre-flight in most of the cell types sampled from the astronauts (Fig. 6c). The ratio of up-regulation in these selected 21 antagomir-rescued genes is more significant than the average up-

regulation ratio of all genes (Fig. 6c, d), suggesting that this subset is more susceptible to expression change due to the space environment.

Cell-dependent differences are detectable for specific genes during spaceflight for the I4 astronauts data (Fig. 6e–n and Supplementary Fig. 6). Two genes, *ARL6IP6* (ADP Ribosylation Factor Like GTPase 6 Interacting Protein 6) and *TSEN15* (TRNA Splicing Endonuclease Subunit 15), are significantly up-regulated post-flight compared to pre-flight in eight of the ten cell types, with CD16+ cells and dendritic cells (DCs) not significantly different for *ARL6IP6* and CD14+ cells, and DCs not significantly different for *TSEN15*. *ARL6IP6* is involved in ischemic strokes in patients[41], which could have implications related to microvessel damage in space. In murine stem cells, *ARL6IP6* exhibits properties of regulating nuclear envelope structure, interacting with proteins involved in nuclear envelope resealing and repair[42]. *TSEN15* is involved in tRNA splicing and intron removal fundamental processes for cell growth and division[43]. It has been implicated in hypoplasia (the impairment in development of tissues and organs due to the decrease in cell number)[44]. Specifically, *TSEN15* has been shown to cause pontocerebellar hypoplasia and progressive

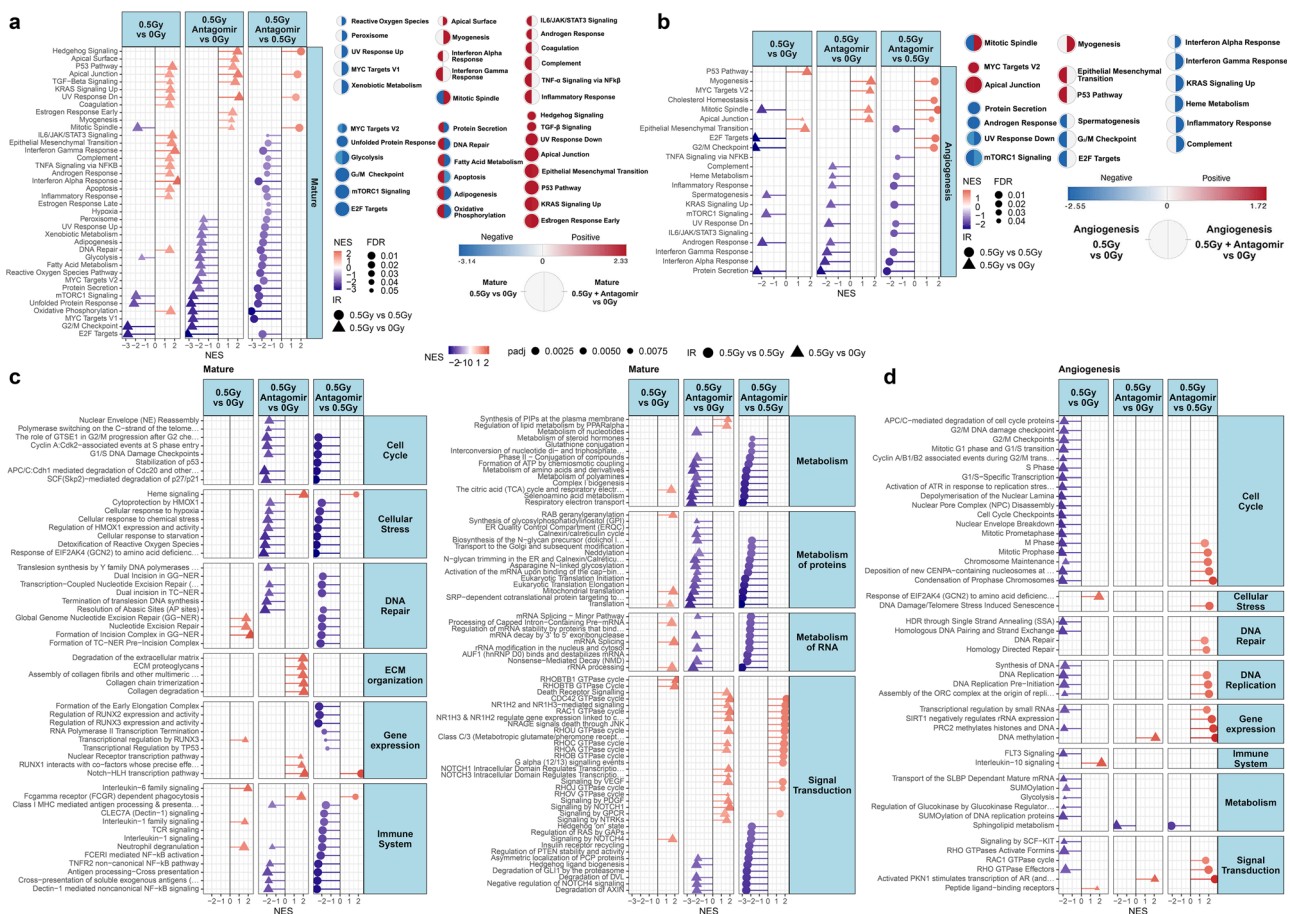

**Fig. 8 | Pathway analysis on the mature and angiogenesis 3D microvessel cell culture models revealing key functions being mitigated by the antagomirs after exposure to GCR irradiation.** Lollipop plots showing Gene Set Enrichment Analysis (GSEA) pathway analysis of the Hallmark pathways for **a** mature and **b** angiogenesis 3D cell culture models. The right of each plot displays circular nodes comparing 0.5 Gy versus 0 Gy (left side of node) with 0.5 Gy plus antagomir treatment versus 0 Gy (right side of node) analyzed with the Cytoscape plugin Enrichment Map. Only significantly regulated pathways with FDR < 0.25 are shown.

GSEA Reactome pathway analysis of RNA-seq data on **c** mature and **d** angiogenesis 3D cell culture models displaying the pathways that are significantly altered by 0.5 Gy irradiation with or without antagomir treatment compared to 0 Gy. Gene sets that were significantly enriched in the same direction for the two treatment groups (both up or both down) are not shown leaving only opposing or single pathway enrichments. Only significantly regulated pathways with FDR < 0.05 are shown. The nominal enrichment score (NES) represents the relative degree a gene set is changed and is corrected for gene set size.

microcephaly. Although this occurs at an early stage of life (*i.e.* infants), one could speculate that TSEN15 may play important roles in cell proliferation in adults. Interestingly, *TSEN15* has also been associated with autoinflammatory neurodegenerative diseases[45]. *ARL6IP6* (ADP ribosylation factor like GTPase 6 interacting protein 6) is predicted to be targeted by miR-125b-5p and let-7a-5p and *TSEN15* is predicted to be targeted by miR-16-5p and let-7a-5 (Fig. 5).

Skin biopsies from the i4 astronauts were also used for RNA-seq comparing post-flight to pre-flight (Fig. 6s) with four different skin regions in the analysis: Outer Epidermis (OE), Inner Epidermis (IE), Outer Dermis (OD), and Vasculature (VA). The VA compartment is closely related to the 3D microvessel cell models which seems to produce the strongest regulation of the 21 key genes. *ARL6IP6* is upregulated in the VA for post-flight compared to pre-flight similar to what is observed in the blood (Fig. 6i–r), while *TSEN15* is down-regulated. Interestingly, 57%, 52%, and 52% of the 21 genes were downregulated for the OE, IE, and VA regions, while for the OD region only 38% were downregulated. Overall, for all the astronaut data, the measured changes of these genes from ex vivo cellular samples may suggest that an interventional treatment with the three antagomirs might lead to improved astronaut health.

## Major cellular pathways altered by radiation exposure and rescued by antagomir treatment

To determine the key pathways that are altered by radiation with or without antagomir treatment, we performed pathway analysis on the bulk RNA-seq data via Gene Set Enrichment Analysis (GSEA). Using the Hallmark database[46], general pathway enrichments were more pronounced in the mature microvessels compared to the angiogenesis model (Fig. 8a, b). Radiation exposure in both these models triggered positive enrichment of pathways typically observed from this treatment such as TP53, Interleukin-6 (IL-6), and TNF-alpha signaling. For both the mature microvessel and angiogenesis cellular models, there was a significantly decreased enrichment associated with the mitotic spindle due to radiation exposure that was reversed by the antagomir treatment. This is consistent with the fact that cellular death following radiation exposure may occur through mitotic catastrophe due to failure of mitotic spindle checkpoints[47].

An increase in inflammatory-related genes and pathways after irradiation is a common occurrence driving the late effects of radiation injury[48,49]. Radiation exposure with antagomir treatment caused a significant reversal in enrichment for several of these pathways. Strikingly, in both mature and angiogenesis microvessels, pathways for

an inflammatory response, TNF-alpha signaling, interferon responses, and IL-6 are significantly decreased by the addition of antagomirs (Fig. 8a, b). Pathways for hypoxia, DNA repair, and oxidative phosphorylation are also activated with the GCR treatment while the antagomir treatment inhibits these pathways. The antagomirs also restored the suppression associated with cell cycle observed with space radiation for the angiogenesis model (Fig. 7b). As a result, the antagomirs appear to be regulating key pathways in the radiation response that may be essential in assisting repair of the damage caused by space radiation in the mature microvasculature.

To further explore the consequences of antagomir treatment to radiation exposure, Reactome genesets consisting of 26 super pathways were examined (Fig. 8c, d). Positively or negatively enriched genesets in the same direction by irradiation with or without antagomirs were excluded. The remaining genesets have opposing enrichments or are enriched by only one treatment. These pathways may represent critical targets that provide beneficial effects from antagomir treatment.

Genes involved in DNA repair by Nucleotide Excision Repair (NER) are enriched in mature microvessels after radiation exposure (Fig. 8c). However, adding antagomirs suppresses these changes as well as changes in genes linked to DNA damage bypass via translation synthesis. This was accompanied by a decrease in pathways involved in cellular stress [e.g. genesets for heme oxygenase 1, hypoxia, detoxification of ROS, EIF2AK4 (GCN2) amino acid deficiencies] as well as a decrease in cell cycle control including telomere maintenance. Notably, there was an overall increase in extracellular matrix (ECM) organization with the irradiation and antagomirs that was not statistically significant with radiation alone. Earlier during the angiogenesis phase of the microvessels, irradiation overwhelmingly decreased cell cycle genesets while the addition of the antagomirs abolished this statistically significant manifestation (Fig. 8d). DNA replication as well as DNA DSB repair by homologous recombination was also returned to non-significant control levels by antagomir treatment.

DNA damage, inflammation, stress and mitochondrial dysfunction are normally associated with the induction of cellular senescence[48]. Therefore, we used publicly available databases compiling markers of cellular senescence[50,51] to investigate whether GCR caused senescence and the antagomirs protected the 3D human microvasculature model from it. Curiously, most senescence pathways were suppressed by GCR, except for oncogene-induced senescence (Supplementary Fig. 8). This is consistent with upregulation of *TP53* target genes, although it dissociates oncogene induction from other causes of senescence induction. The antagomirs did not revert these phenotypes, except in the angiogenesis model, where they promoted cellular senescence in GCR-exposed cells.

## Antagomir treatment after space radiation modulates critical cellular pathways

To specifically observe how key immune and inflammatory pathways are regulated with and without the antagomir treatment in the space environment, we analyzed a custom curation of genesets for innate and adaptive immune pathways (Fig. 9). There was an overall increased activity with GCR radiation exposure for the mature microvessel culture. However, when the antagomirs were applied, general inflammatory and immune pathways were restored back to the 0 Gy level for the mature microvessel model with a negative regulation in the angiogenesis cell model indicating that this countermeasure treatment diminished this response of GCR irradiation in the mature model (Fig. 9a).

In mature 3D microvessel models, GCR radiation treatment significantly upregulated several interferon-driven sensitive genes (ISGs), including *DDX58, IFIH1, IFIT2,* and to a lesser extent, *IFIH1, IFIT1, IFIT2, IFIT3, IFIT5, IFITM1, IFITM3, OAS1-2* amongst others (Fig. 9b). Also upregulated by GCR radiation treatment were the cytokine/chemokine genes *CXCL10* and *CXCL11*, which function as pro-inflammatory molecules associated with activation, differentiation, and chemotaxis of immune cells. In contrast, treatment with the antagomirs produced decreased levels of *CXCL10* and *CXCL11* and of all these ISG genes except for *IFIH1* and *OAS2*.

The *NLRP3* inflammasome is a multiprotein complex formed by *NLRP3, PYCARD* and *CASP1* that plays a pivotal role in regulating the innate immune system and inflammatory signaling. Upon activation by cytosolic danger signals, the *NLRP3* inflammasome activates *CASP1* triggering the maturation of *IL-1β, NF-κB* activation, and the initiation of cell death via *GSDMB*[52]. GCR-treated cell cultures significantly upregulated *NLRP3* and, to a lesser extent *PYCARD, CASP1,* and *NFKB1,* while treatment with antagomirs lessened the induction of these genes (Fig. 9b). Together, the increases in ISGs, cytokines/chemokines and *NLRP3* inflammasome indicate that GCR radiation exposure promotes a robust innate immune response associated with increased production of pro-inflammatory molecules and immune cell activation. In contrast, the antagomir treatment downregulates and restores these factors back to normal levels. The adaptive immune genes were not significantly altered, consistent with the transient cell stress induced by the GCR treatment and potentially due to the fact that the 3D cell model does not contain any T cells (Fig. 9c).

## Mitochondrial damage mitigation after antagomir treatment with space radiation

Mitochondrial stress has been previously implicated as a critical and consistently dysregulated phenotype during spaceflight[12]. Mitochondria play a crucial role in metabolism by integrating signals from stress and the environment, such as nutrient deprivation and oxidative stress, to coordinate cellular metabolism[53]. Thus, the MitoPathways collection[54] was used to examine mitochondria-related activity in our 3D cell culture model. GCR exposure combined with antagomir treatment in the mature microvessels significantly decreased expression of 106 of 149 available genesets (Fig. 10a). Radiation exposure alone did not significantly alter 72 of these genesets as highlighted in Fig. 10b.

The largest of six major networks involves metabolism-linked MitoPathways where antagomirs with GCR exposure overwhelmingly downregulated 44 of 50 genesets (Fig. 10b). Negative enrichment in 8 of 9 subpathways are associated with carbohydrate, lipid, amino acid, nucleotide, metal, vitamin, and sulfur metabolism as well as detoxification. The substantial metabolic alterations may depend on the changes affecting the mitochondria which were also largely downregulated indicating an overall decrease in cellular energetics. The OXPHOS and Mitochondrial Central Dogma MitoPathways include crucial factors for oxidative phosphorylation functions (Complexes I-V subunits and assembly factors) and proteins encoded within mitochondrial DNA and factors involved in their expression, respectively. These genesets were nearly all decreased when antagomirs were added to the radiation exposure. The increased enrichment of 7 subpathways by radiation treatment alone (OXPHOS and OXPHOS subunits, CI and CV subunits, Complex I, Mitochondrial Ribosome, and Translation) was completely reversed with the antagomirs.

In response to stress stimuli, eukaryotic cells activate the integrated stress response (ISR) to restore cellular homeostasis via communication between organelles[55]. The communication between the mitochondria and the nucleus is essential for cellular homeostasis[42], due to the large number of mitochondrial proteins encoded by the nucleus. Considering the observed decrease in mitochondrial and changes in stress pathways, the ISR genes were examined in more detail. GCR exposure in the mature 3D cell cultures downregulated ISR genes involved in amino acid uptake/biosynthesis, death factors, endoplasmic reticulum unfolded protein response (ER-UPR) activation, and lipid metabolism, while only ISR survival genes were

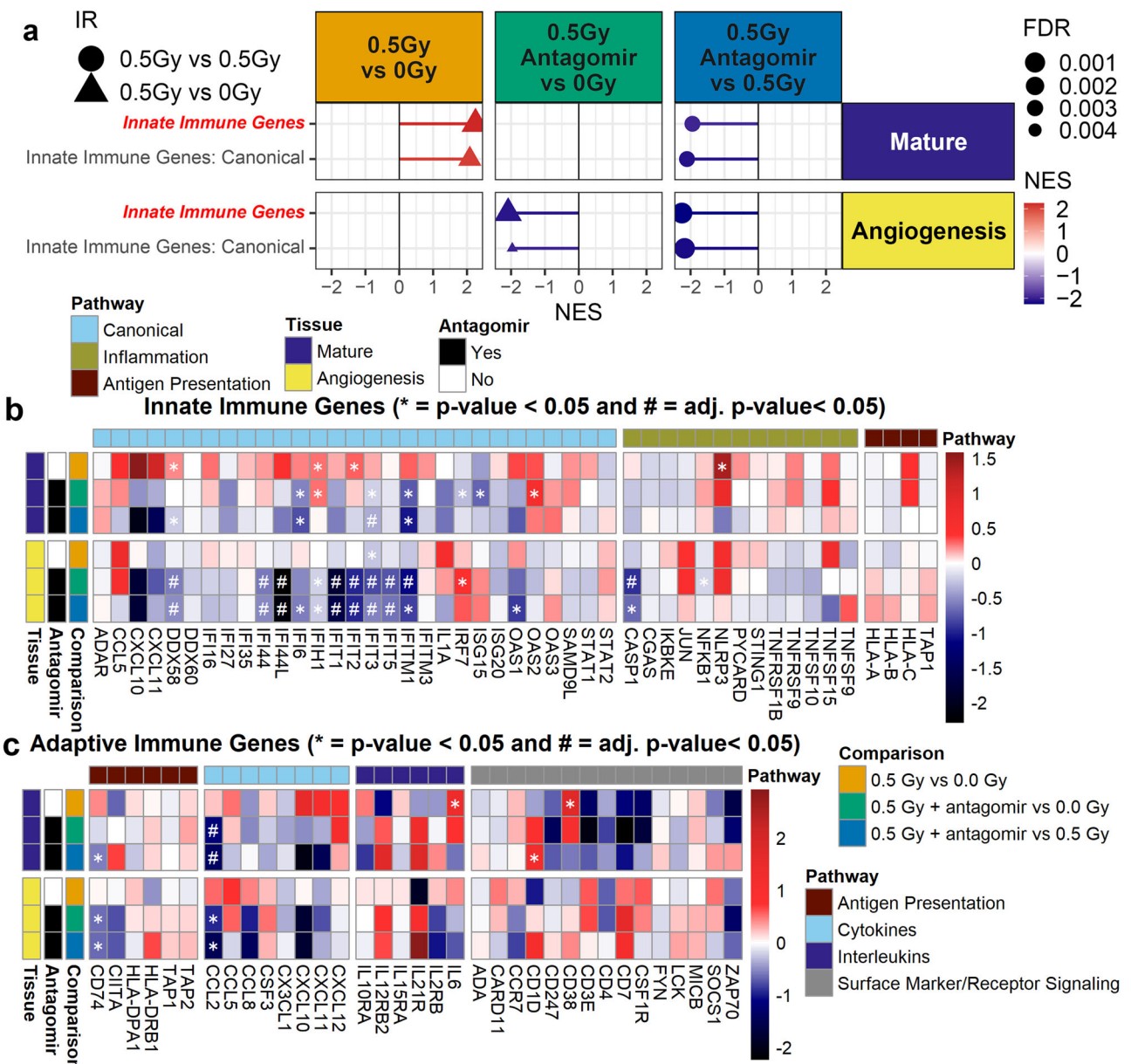

**Fig. 9 | Antagomir countermeasure mitigates increases in innate immune activity. a** Lollipop plot of GSEA analysis on RNA-seq data using a custom curation of innate and adaptive immune pathways for mature and angiogenesis 3D microvessel cell culture models. The red italic bold font names indicate the primary pathway containing all genes from the sub-pathways. Only pathways with a FDR < 0.25 are shown. Heatmaps of the t-scores for the individual genes in the **b** innate and **c** adaptive immune custom pathways. (**p*-value < 0.05). For all RNA-seq data Wald test and the likelihood ratio test was used to generate the F statistic *p*-value.

positivity enriched (Fig. 10c). Fewer genesets were affected by GCR exposure, which increased the ATF4 targets geneset and decreased ER-homeostasis. Initiation of the ISR is driven by the activation of the kinases HRI (EIF2AK1), PKR (EIF2AK2), PERK (EIF2AK3), or GCN2 (EIK2AK4) which occurs in the mature cell model with GCR radiation (Fig. 10d). These kinases phosphorylate eIF2α, impairing cytosolic protein synthesis and activating expression of the ISR transcription factors *ATF4, ATF5,* and *CHOP*. Increased levels of *ATF4, ATF5,* and *CHOP* in the angiogenesis cell model (Fig. 10c, d) induce the expression of the downstream ISR-target genes involved in redox maintenance, lipid metabolism, amino acid uptake/biosynthesis, ER-homeostasis, death, survival, and the ER and mitochondrial UPR. For irradiation combined with the antagomirs, genesets are returned to baseline. These results suggest that cells activate ISR as an adaptive cellular response after exposure to space radiation, whereas this activation is prevented by the antagomirs (Fig. 10d).

## Discussion

Our previous study[13–15] demonstrated a significant association between circulating miRNAs and spaceflight-associated health risks. These 13 circulating miRNAs were connected to the immune system, cellular differentiation, motility, proliferation, and survival[13,15]. Three miRNAs, miR-16-5p, miR-125b-5p, and let-7a-5p, were linked to an increased risk of cardiovascular disease during spaceflight[13,15,56]. A collapse of microvessels in 3D cell cultures after exposure to space radiation[13,21] was associated with breakdown of the endothelial barrier and the loss of tight junctions[57]. GCR-induced inhibition of microvessels during angiogenesis was dependent on the linear energy transfer (LET) of the charged particles[58,59]. Although the microvessel cells are post-mitotic, differential effects on DNA repair occur according to radiation quality (e.g. LET)[20]. The mixed simplified simulation of GCR uses five ions of different energies (Hydrogen, Helium, Oxygen, Silicon, and Iron)[48] which are highly effective at inhibiting angiogenesis[14].

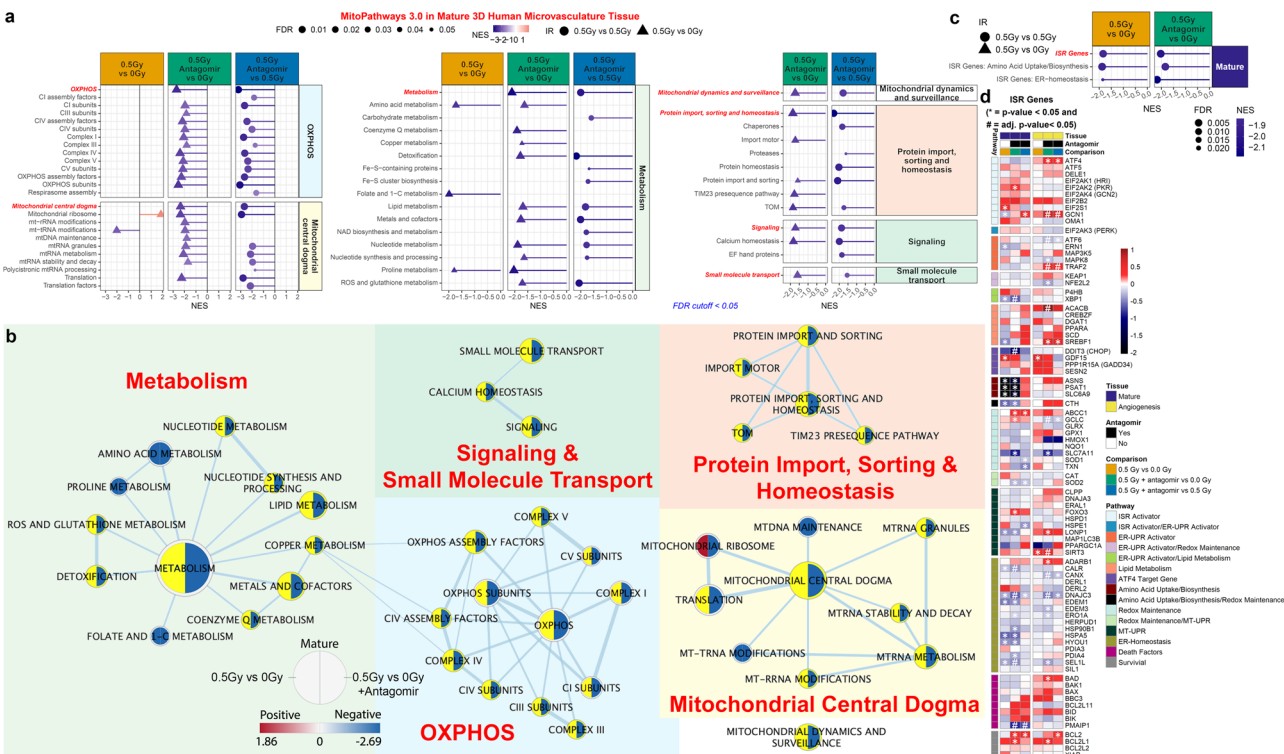

**Fig. 10 | Mitigation of mitochondrial dysregulation and integrated stress response (ISR) with antagomir treatment on mature and angiogenesis 3D cell culture models irradiated with GCR. a** Lollipop plots for the GSEA analysis on mitochondrial pathways determined from MitoCarta/MitoPathway for 3D mature microvessel cell culture (FDR < 0.25). The red italic bold font names indicate the main general pathway containing all the genes from the sub-pathways. **b** Network representation analyzed with Cytoscape for the mitochondrial pathways from MitoCarta/MitoPathway. The node size indicates the number of significantly regulated genes associated with each pathway. Each node shows the pathway regulation with FDR < 0.25 for 0.5 Gy versus 0 Gy (left side) and 0.5 Gy with the antagomirs versus 0 Gy (right side). Genesets not significantly expressed by 0.5 Gy versus 0 Gy are highlighted in yellow. **c** GSEA analysis on the custom integrated stress response (ISR) pathways for mature and angiogenesis RNA-seq data. The red italic bold font names indicate the main general pathway containing all the genes from the sub-pathways. Only significantly regulated with pathways with FDR < 0.25 are shown. **d** Heatmaps of the t-scores for the individual genes in the ISR custom pathways. (*$p$-value < 0.05). For all RNA-seq data Wald test and the likelihood ratio test was used to generate the F statistic $p$-value.

Several studies have shown the cellular consequences of miRNA activity, including the three miRNAs researched in the present study. *Let-7a* is involved in many cellular processes, immunity, and protective functions[60]. The majority of cellular processes involving *let-7a* are well studied in development, proliferation, differentiation, and cancer, but its role in inflammation, cardiovascular diseases, and/or stroke are not well understood[61]. *Let-7a* is involved in hematopoietic stem and progenitor cell homeostasis such as self-renewal, proliferation, quiescence, and differentiation by blocking the TGFβ pathway and amplifying Wnt signaling[62].

MicroRNA 16 (miR-16) is linked to human malignancies[63] and can modulate the cell cycle, inhibit cell proliferation, promote cellular apoptosis, and suppress tumorigenicity[64]. The targets of miR-16 may be directly responsible for these effects such as the anti-apoptotic gene *BCL-2* (B-cell lymphoma 2)[65], numerous genes involved in the G1-S transition, such as cyclin D1, cyclin D3, cyclin E1 and *CDK6* (cyclin-dependent kinase 6), and genes involved in the Wnt signaling pathway, such as *WNT3A* (wingless-type MMTV integration site family, member 3A)[66]. In endothelial cells, angiogenic factors, including vascular endothelial growth factor (*VEGF*), *VEGF* receptor-2 (*VEGFR2*), and fibroblast growth factor receptor-1 (*FGFR1*), were identified by bioinformatic approaches and subsequently validated as targets by miR-16[67].

The miR-125 family plays an important role in the growth and development of animals, as well as stress response, p53 activation[68] and the occurrence and development of cancer[69]. In addition, the role of miR-125 in cardiovascular and cerebrovascular diseases cannot be ignored. The miR-125 family is associated with the development and

differentiation of mammalian embryonic hearts[70]. Furthermore, it plays an important role in diseases and pathophysiological processes such as, coronary heart disease, myocardial infarction, ischemia-reperfusion injury, stroke, myocardial fibrosis, endothelial cell injury and myocardial cell apoptosis[70]. A recent paper has described the presence of miR-125b in osteoblast-derived matrix vesicles (MVs)[71] and in bone matrix where it hampers osteoclastogenesis and bone resorption by targeting PRDM1 (PR domain zinc finger protein 1), thus suggesting that miR-125b might have an active role in the development of vascular calcifications[72].

All three miRNAs were initially found to be associated with cancer and neoplasia. However, unregulated cell cycle control is unlikely to be the only pathway that mitigates the biological effects of GCR. Our results from the mature microvessel model support this notion, since mature microvessels are postmitotic and the timing of events is too rapid to be downstream of cell cycle control, although this does not rule out an indirect involvement of cell cycle proteins. At 24 h after GCR irradiation and 48 h after antagomir treatment, protection of microvessel morphology is observed compared to the GCR-stressed control not treated with antagomirs.

Individual inhibition of each target miRNA and the resulting mitigation of GCR effects suggests that each miRNA is similarly effective and therefore the targets of each must contain genes that are active in this process. Twelve of 21 key genes targeted by at least two of the three miRNAs (Fig. 5a) are potential candidates for protecting the microvessel physiology. These include genes involved in apoptosis and p53-mediated cell death, cell cycle and mitotic phase transitions, innate immune response, and mitochondrial processes related with

amino acids, and cholesterol metabolism (Supplementary Fig. 5). However, apoptosis is not measurable in these cultures as much higher doses of heavy and light ions are needed to detect this mechanism[21] by TUNEL assay although this does not rule out earlier apoptosis signaling. Direct action of genes involved in the cell cycle and mitotic phase transitions is unlikely in mature microvessels. Involvement of cellular programs from the innate immune response is a possibility, since immune-associated genes seem to be heavily regulated by space radiation and miRNAs. The effects of the antagomir cocktail on suppressing DNA repair indicates a much earlier time of action, likely upstream of the rescue of angiogenesis or the protection of mature vessels. This suggests that early response genes involved in radiation exposure might be targets for the antagomirs; these would include genes of general inflammatory and immune pathways. There are 5 genes that are affected in common with all 3 antagomirs. When inhibited, all products of these genes are candidates for mediating the resistance to radiation damage, although the effects of other genes affected by each individual antagomir cannot be ruled out. Each gene has functions that could contribute to GCR resistance. *APAF1* and *NIBAN1* are involved in the control of apoptosis and may prevent progression towards senescence[73]. *XPO6* and *SLC7A1* are involved in transport across membranes between cell compartments, a process involved in early responses to cell stressors[74,75]. *PDPR* regulates pyruvate phosphorylation and may facilitate protective functions that require a higher energy output[76]. Overall it appears that several pathways are involved in protection and are acting in concert to push the cells into a more healthy and active biological state.

Changes in immune response appear to be a hallmark of space radiation and targeted by miRNAs. We previously have shown that cytotoxic and helper T cells are suppressed and key innate immune response-related factors are dysregulated in C57BL/6 female mice exposed to 0.5 Gy GCR irradiation[14]. Markers of CD4 (helper) and CD8 (cytotoxic) T cell populations in the I4 astronauts (Fig. 6j, k) also indicate dysregulation occurring during spaceflight. The similarities between both in vivo and in vitro experiments demonstrate a parallelism that gives confidence about using the 3D cell culture model to confirm the effectiveness of the antagomir treatment strategy to prevent and countermeasure space radiation damage.

Spaceflight can affect mitochondrial stability[12]. The ability of antagomirs to rescue mitochondria-associated pathways affected by radiation indicates the potential for using these antagomirs to rescue harmful phenotypes in space travelers. In particular, the OXPHOS pathway, which was downregulated upon treatment with GCR-stress, showed a rescue-like effect (Fig. 10). Impaired OXPHOS can lead to cell death due to insufficient ATP for energy use, changes in mitochondrial membrane potential or overproduction of prooxidant agents[77]. Electron carriers, like the amidoxime-reducing complex, assists in this process and show downregulation in the presence of radiation. This radiation phenotype has the potential to progress to inflammation. Specifically, the genes whose upregulation was returned to normal by the addition of antagomir include those of complex I, complex V, and other OXPHOS subunits.

Genotoxic stress caused by ionizing or UV radiations is a potent inducer of cellular senescence. Curiously, GCR suppresses most of the pathways associated with cellular senescence curated based on data from fibroblasts, renal epithelial cells and adipocytes induced to senescence by different stimuli, including X-ray irradiation, oncogene overexpression, antiretroviral drug or hyperinsulinemia[50,51]. This suggests that either GCR does suppress classical cellular senescence or GCR-induced senescence in our 3D cell culture differs from the previously studied mechanisms of senescence induction by radiation (Supplementary Fig. 8). The latter is supported by previous evidence demonstrating that different senescence stimuli drive very distinct phenotypes[51]. The only senescence-related pathway induced by GCR in our model is oncogene-induced senescence, which is consistent with

activation of p53 target genes. The antagomirs have no effect on the mature culture model, but in the irradiated angiogenesis model they promote various senescence-related pathways. Future studies are required to assess the role of these miRNAs in senescence induction by more classical stimuli. Future studies are also required, as stated above, to decipher GCR effect on multicellular microvascular models, in which differential irradiation effect on the different cell population (fibroblast/pericyte and endothelial cells for example) might impaired, or not, paracrine communication and then the global effect in vivo (human).

The interplay between orphan genes and miRNAs has not been previously studied. We observe a significant change in orphan gene expression from GCR irradiation stress and from antagomir treatment (Fig. 4d and Supplementary Fig. 4). Many orphan genes were upregulated in the 3D mature microvessel cell culture and interestingly, downregulated in the angiogenesis model. Some of these orphan genes may be integrated into the metabolic and regulatory processes that are targeted by the miRNAs. Orphan genes themselves are simple proteins that often act by binding to conserved proteins such as transcription factors or receptors[31,78]. Some orphan genes are key for reproduction, some are toxins, and many confer resistance to stress[30,31,79–82], however, orphan genes do not appear to encode enzymes, likely because of the very long time scale required for evolution of a specialized catalytic site[82]. Many orphan genes are expressed to high levels only in discrete tissues or under specific abiotic stresses or diseases[30,32,79,82,83]. The alteration of orphan gene expression with GCR irradiation and antagomir treatment warrants further examination.

Due to the nature of performing experiments in the space environment, it can be limiting and difficult to simulate perfectly the conditions in space in ground experiments. So to perfectly recreate the space radiation dose and dose-rate that astronauts will receive during long-duration spaceflights (lasting several months) is not possible. The ground-based experimental setup reported in this manuscript used for irradiating with GCRs currently represents the best approximation for validation of predictive models and while enabling countermeasure developments. Thus, extrapolation and comparison from the astronaut data to cell culture studies conducted on Earth as well as mice experiments both on Earth and during spaceflight are key to meaningful interpretation of biological results applicable to space flight. Lastly, it is possible that the differences in the radiation delivery might account for any deviations between the results we are observing.

The present microvascularized model is a simplified system hosting endothelial cells only and one shall point out the absence of the very important pericytes. Indeed, it is well known that when endothelial cells organize into microvascular networks (during angiogenesis), stromal cells associate with newly formed microvessels to stabilize the structure. Fibroblasts can play this role by adopting a pericytic-type phenotype[84–86]. To push forward the full understanding of the GCR effect on angiogenesis, it might be necessary, in a close future, to complexify the model, at least toward a multicellular compartment[87]. In addition, it will be essential for future studies to test the toxicity of the antagomirs in in vivo models, since the tissue model utilized for this manuscript is limited for this purpose.

The inhibition of three miRNAs, miR-16-5p, miR-125b-5p, and let-7a-5p, facilitates the reduction of cellular damage from ionizing radiation exposure. Treatment with antagomirs that inhibited these three miRNAs before and after radiation exposure in mature and angiogenesis 3D human HUVEC cell culture models, completely prevented the microvessel collapse and inhibition of microvessel development caused by GCR exposure. The potential effectiveness of these antagomirs to reduce prolonged tissue damage through modified DNA repair activity and alterations in cellular mitochondria offers a unique target for the development of radiation countermeasures for spaceflight as well as for terrestrial applications.

## Methods

### Ethical statement

All mice experiments were approved by Brookhaven National Laboratory's (BNL) Institutional Animal Care and Use Committee (IACUC) (protocol number: 506) and were performed by trained personnel in AAALAC accredited animal facilities at BNL, while conforming to the U.S. National Institutes of Health Guide for the Care and Use of Laboratory Animals. All methods were carried out in accordance with the relevant guidelines and regulations and are reported in accordance with ARRIVE guidelines.

All human studies were done with ethical approvals with established and approved IRBs and according to the criteria set by the Declaration of Helsinki. The Inspiration4 (I4) mission was performed at Weill Cornell Medicine. Blood samples were provided by SpaceX Inspiration4 crew members after informed consent for research use. The procedure followed guidelines set by Health Insurance Portability and Accountability Act (HIPAA) and operated under Institutional Review Board (IRB) approved protocols and informed consent was obtained. Experiments were conducted in accordance with local regulations and with the approval of the IRB at the Weill Cornell Medicine (IRB #21-05023569). All crewmembers consented to data and sample sharing.

The NASA Twin Study subjects signed informed consent according to the declaration of Helsinki and 14 CFR Part 1230 for collection and use of sample materials and data in research protocols at NASA and the collaborating institutions. Study protocols were approved by the NASA Flight Institutional Review Board (protocol number Pro1245) and all participating institutions. Also the data is hosted at the NASA Life Sciences Data Archive (LSDA). Informed consent was provided by all participants in the NASA Twin Study. The JAXA astronaut data was obtained from NASA's GeneLab Platform and this study used only published aggregated quantification values.

### Cell culture

Prior to culturing 3D microvessels, HUVEC (Lonza Inc, Allendale, NJ, USA), primary cells isolated from the vein of the umbilical cord were cultured as 2D monolayers in complete endothelial basal medium (EBM; Lonza Inc., Allendale, NJ, USA), containing EGM medium (serum free) supplemented with 2% fetal bovine serum (FBS), human epidermal growth factor, hydrocortisone, and bovine brain extract. Cells were kept in a humidified incubator (5% CO$_2$, 95% air) and the medium was changed twice a week. All cells used for experiments were cells thawed from a fresh vial.

### 3D HUVEC vessel cell model

In this study, 3D cell culture was performed according to the method initially described by Davis et al.[88] with modifications described in Grabham et al.[21]. Briefly, a collagen gel solution (Rat tail Collagen Type 1, BD Biosciences, Bedford MA) was prepared on ice by mixing together the following stock solutions: 0.35% collagen solution, 10 Å- M199 medium, and 1 M HEPES (pH 7.4) in a ratio of 8:1:1 by (volume). This solution was then mixed with matrigel (BD Biosciences, Bedford, MA) in a ratio of 3:1 by (volume). HUVECs grown in EGM on 2D cell culture dishes (80–90% confluence) were detached with a trypsin solution (0.025%) and resuspended in EGM-2 (Lonza Inc, Allendale, NJ, USA). The cell solution was mixed with the gel solution in a ratio 1:5 (by volume). The resulting cell suspension contained a final cell suspension of 1×10$^6$ cells/ml. 25 µl was dropped onto the cell growing surface inside a tissue culture flask (T25) (Corning) and allowed to gel at 37 °C for 30 min. The gel matrices were then overlaid with EGM-2 medium for 7 days with daily media changes. EBM-2 consists of EGM-2 media supplemented with 2% FBS, human fibroblast growth factor B (hFGF-B), human epidermal growth factor (hEGF), human vascular endothelial cell growth factor (hVEGF), long R insulin-like growth factor 1 (R3-IGF-1), ascorbic acid, hydrocortisone and heparin. 50 nM Phorbol 12- Myristate 13-acetate (PMA) (Millipore Sigma) was also added to the growth media. Cells embedded in 200–300-µm thick gel matrices were incubated at 37 °C in a humidified incubator (5% CO$_2$, 95% air) for the times indicated. The medium was refreshed every 24 h or as otherwise stated in individual experiments. For mature microvessels, cells were cultured for 6 days and treated with AUMantagomir™ (AUM biotech) (0.5 µM each) 24 h before radiation exposure then fixed 24 h later. For the angiogenesis model, cultures were irradiated on day one and cultured for 6 more days before fixation. Preliminary experiments showed that antagomir treatment on days 2 and 3 were most effective. For DNA damage/repair (i.e. 53BP1 staining), cultures were fixed at 90 min after irradiations.

### Simulated Galactic Cosmic Radiation (GCR) exposure

Irradiation of microvessel models and animals were conducted at the NASA Space Radiation Laboratory (NSRL) at Brookhaven National Laboratory (BNL, Upton, NY). Samples were positioned in the plateau region of the Bragg curve and irradiated at room temperature. Dosimetry was performed by the NSRL physics staff. For microvessel models, cultures were seeded at Columbia University in New York City then transported to BNL the same day. Irradiations were carried out the following day (Day 1) at NSRL. A 20 cm × 20 cm beam was utilized. Since the heavy-ion beam at NSRL is horizontal, flasks containing 5 ml of medium were upended to a vertical position for a few minutes during irradiation. All cultures were irradiated with 0.5 Gy of simplified simulated GCR (referred to as GCR). The irradiation was used with ions, energy, and doses determined by a NASA consensus formula for 5 ions: protons at 1000 MeV, $^{28}$Si at 600 MeV/n, $^4$He at 250 MeV/n, $^{16}$O at 350 MeV/n, $^{56}$Fe at 600 MeV/n, and protons at 250 MeV, in the following proportions – 1000 MeV protons at 34.8%, 250 MeV protons at 39.3%, $^{28}$Si at 1.1%, $^4$He at 18%, $^{16}$O at 5.8%, and $^{56}$Fe at 1%. This mixture is the simplified version of the full GCR simulation and represents the proportions of ions found in space and thus translates to exploratory class missions[48]. The low LET particles (Protons and Helium) make up the majority of charged particles although high LET ions generally have a greater relative biological effect (RBE).

### Quantification of 3D human cell culture models after irradiation: Immunocytochemistry, Imaging, and microvessel analysis

All microvessel cultures were fixed and stained as for monolayers. Fixed in 4% paraformaldehyde followed by three washes in Phosphate buffered saline (PBS) with 0.5% Triton-X, and finally in PBS. 5-(4,6-Dichlorotriazinyl) Aminofluorescein (DTAF) (Thermofisher Scientific) was used to as a fluorescent green stain for all proteins to visualize microvessels (Invitrogen, Carlsbad, CA, USA). 53BP1 foci were immunostained using a polyclonal antibody against 53BP1 (Catalog #: NB100-304, Novus Biologicals, Littleton, CO, USA) and a counterstain for nuclei with YOYO green (Invitrogen, Carlsbad, CA, USA) followed by Alexa fluor conjugates 495 or 488 (Invitrogen, Carlsbad, CA, USA). Images were captured on a Nikon TE 200 confocal C1 microscope with EZ-C1 software. Analysis of images was carried out using NIH Image J v. 1.44k software[68]. Initial analysis on microvessels was carried out on randomly selected 20X fields from within the gel matrix. Thresholded images of microvessels were outlined as perimeters then divided by 2 for length measurements. For the improved analysis the field size was increased 4-fold by using images captured by a 10× objective. The mature microvessels were defined more rigorously as those of a minimum diameter of 12.5 µm. This step eliminated many immature thin vessels and reduced the effective doses. Finally, the sample fields were standardized, by imaging microvessels at a constant depth (50 mm above the substrate). All analyses were carried out blind. DNA double strand break repair foci assay was carried out by counting the number of foci per nucleus.

## MicroRNA predictions of DNA DSB gene targets and network generation

We predicted specific DNA DSB genes that are targeted by three miRNAs: miR-16-5p, miR-125b-5p, and let-7a-5p. The DNA DSB gene list was determined by filtering out all DNA DSB related Gene Ontology (GO) pathways[46,89]. Each GO DNA DSB pathway will have curated genes associated with the pathway. We compiled all genes related to these into a master list and deleted any duplicated genes. We then utilized the ClueGO+CluePedia plugin[90,91] in Cytoscape[92] to map the relationship between the gene targets for the three miRNAs. This was done by including all DNA DSB genes and the three miRNAs in the initial query list. The edges and connections were made with CluePedia utilizing the following miRNA-mRNA databases in the plugin: CluePedia STRING-Actions (v.11.0_9606_27.02.2019), CluePedia microRNA.org-human predictions_s_O_aug2020, miRDB_v6.0, miRanda-miRNAs v5-2012-07-19, mirTarBase, and mirecords.umn.edu.validated.miRNAs-2010-11-25.

## Nucleic acid extraction for 3D human cell culture models

We used $n = 5$ biological replicates for each condition for RNA-seq. RNA was extracted using the RNeasy Universal kit (Qiagen, Valencia, CA). Gel matrix cryotubes containing the cells of interest were taken out of −80 °C and 1 mL of QIAzol reagent was added to the tubes. Each sample contained 300–800 μL of gel matrix and approximately 500,000 cells. While the gel matrix thawed in QIAzol, the solution was pipetted and vortexed until fully dissolved. The lysate was passed through a QIAshredder column (Qiagen, Valencia, CA) to further homogenize the cells. 100 μL of gDNA eliminator solution was added. The resulting lysate from each sample was then used to isolate and purify RNA following the manufacturer protocol. RNA was eluted in 30–50 μL RNAse free H2O. Concentration of all RNA samples were measured using Qubit 3.0 Fluorometer (Thermo Fisher Scientific, Waltham, MA) with Qubit RNA BR kit following SOP 4.1-RNA/DNA/miRNA/cDNA Quantification using Qubit Fluorimeter[93]. RNA quality was assessed using the Agilent 4200 TapeStation with the Agilent RNA ScreenTape (Agilent Technologies, Santa Clara, CA).

## Spike in Protocols before library preparation in 3D cultures

ERCC ExFold RNA Spike-In Mixes (Thermo Fisher Scientific, Waltham, MA, Cat 4456739, v92), Mix 1 or Mix 2, were added on the day of library prep at the concentrations suggested by the manufacturer protocol. SOP 5.2 Use of ERCC Spike-In Mixes and UMRR/UHRR Controls for Total RNA Sequencing[93].

## Library construction for RNA-sequencing on 3D cultures

RNA ribo-depletion and library preparation was done with Illumina TruSeq Stranded Total RNA Library Prep Gold (Illumina Inc., San Diego, CA). Input RNA were approximately 500 ng with RIN > 6.5 (average RIN of 8.7). Index adapters used were at 1.5 μM (IDT, 384-well xGen Dual Index UMI Adapters). 15 PCR cycles were performed. Library fragment size was assessed using Agilent 4200 TapeStation with D1000 DNA ScreenTape (Agilent Technologies, Santa Clara, CA) following GeneLab SOP 6.3 Quality analysis of sequencing libraries using 4200 TapeStation System with D1000 reagent kit[93]. Pooled library concentration was measured by Qubit 4 Fluorometer (ThermoFisher Scientific, Waltham, MA) following SOP 4.1-RNA/DNA/miRNA/cDNA Quantification using Qubit Fluorimeter. Library quality assessment was performed on iSeq100 (Illumina, San Diego, CA).

## Bulk RNA-sequencing and data processing on 3D cultures

RNA sequencing was performed by GeneLab Sample Processing Lab on Illumina NovaSeq 6000. Sequencing was set up as follows: Read 1: 151 bp, Index1: 17 bp(8 bp index +9 bp UMI), Index2: 8 bp Read 2: 151 bp. PhiX was included as an internal control and to increase library diversity. Protocol followed is SOP 7.1-Setting up NovaSeq 6000 and iSeq 100 Sequencers[93].

Raw fastq files were assessed for percent rRNA using HTStream SeqScreener (version 1.3.2) and filtered using Trim Galore! (version 0.6.7) powered by Cutadapt (version 3.7). Raw and trimmed fastq file quality was evaluated with FastQC (version 0.11.9), and MultiQC (version 1.12) was used to generate MultiQC reports. *Homo sapiens* STAR and RSEM references were built using STAR (version 2.7.10a) and RSEM (version 1.3.1), respectively, Ensembl release 101, genome version GRCh38 (Homo_sapiens.GRCh38.dna.primary_assembly.fa) concatenated with ERCC92.fa from ThermoFisher (https://assets.thermofisher.com/TFS-Assets/LSG/manuals/ERCC92.zip), and the following gtf annotation file: Homo_sapiens.GRCh38.101.gtf concatenated with the ERCC92.gtf file from ThermoFisher (https://assets.thermofisher.com/TFS-Assets/LSG/manuals/ERCC92.zip). Trimmed reads were aligned to the *Homo sapiens* + ERCC STAR reference with STAR (version 2.7.10a) and aligned reads were assessed for strandedness using RSeQC Infer Experiment (version 4.0.0) then aligned reads from all samples were quantified using RSEM (version 1.3.1), with strandedness set to forward. pyrpipe was used for quantification of the orphan genes and other EB genes[23,69]. Using DESeq2, the likelihood ratio test (LRT) was performed separately for the 3 conditions on each 3D microvessel cell culture model. Quantification data was imported to R (version 4.1.2) with tximport (version 1.22.0) and normalized with DESeq2 (version 1.34.0) median of ratios method. The data was normalized twice, each time using a different size factor. The first used non-ERCC genes for size factor estimation, and the second used only ERCC group B genes to estimate the size factor. Both sets of normalized gene counts were subject to differential expression analysis. Differential expression analysis was performed in R (version 4.1.2) using DESeq2 (version 1.34.0); all groups were compared using the log2(FoldChange) values and the Wald test and the likelihood ratio test was used to generate the F statistic $p$-value. Gene annotations were assigned using the following Bioconductor and annotation packages: STRINGdb (v2.8.4), PANTHER.db (v1.0.11), and org.Hs.eg.db (v3.15.0). We have utilized for our analysis a $p$-value $< 0.05$ for statistical significance and acknowledge the potential of more false positives to occur. All TPM values can be found in Supplemental Data 1.

Raw RNA-seq data is available on the NASA OSDR's Biological Data Management Environment[94] with accession number: OSD-577.

## Determination of the 21 key gene targets for the three miRNAs, cumulative plots for the miRNA targets, and statistical test

To determine the 21 key gene targets for miR-16-5p, miR-125b-5p, and let-7a-5p, we found the genes that were statistically significantly downregulated for 0.5 Gy vs 0 Gy and no longer significant for 0.5 Gy with antagomir vs 0 Gy. We then utilized mirDIP[95] to determine the gene targets for all 3 miRNAs with a criteria of medium score class. The medium score class has been shown in the literature to provide a reasonable criteria for determining gene targets for miRNAs[35]. We overlapped these two lists of genes and found 21 genes. The 21 gene fold-change values, statistics, and TPM values can be found in Supplemental Data 2.

We used the "Target Mining" function of the online platform miRwalk[96]. It allowed searching the 21 gene targets against the three miRNAs (hsa-miR-125b, hsa-let-7a and hsa-miR-16), and investigate their 3′-, 5′-UTR and coding sequence (CDS) regions with a binding probability cutoff of at least 0.75 (Sankey plot part of Fig. 5b–d). To this, we have added the genes' respective log2 fold change given a GCR (0.5 Gy) and Sham control (0 Gy).

To build the cumulative plots for the 21 miRNA gene targets, we obtained the information about whether these genes have 7mer or 8mer sites that match the seed region of each the three miRNAs (hsa-miR-125b, hsa-let-7a and hsa-miR-16) using the TargetScan v8.0 database[97]. mRNAs were then grouped according to miRNA seed sequences (7mer-1A, 7mer-m8 or 8mer or 'no site'), and the cumulative plot was generated using the mRNA log2 fold change values using the

*ecdfplot* method from seaborn[98]. To compare distributions of mRNA fold change values for each seed match with the 'no site' case, we calculated the Kolmogorov–Smirnov test using the *kstest* function from scipy in python[99].

To determine the experimental validation for the 21 gene targets to the miRNAs, we utilized three databases which are: miRTarBase v9.0[37], TarBase v9.0[38], and StarBase v2.0[39]. Both TarBase and miRTar-Base include literature references for the experimental validation. We also provided additional literature references for some genes as indicated in Supplementary Table 1. We displayed these results both in Supplementary Table 1 and as bar plots (Fig. 5j) using ggplot2 (ver 3.3.5).

### Conserved miRNA analysis between humans and mice
Selected precursor and mature miRNA sequences from human and mouse were extracted from miRBase v.22.1[100]. After BLASTN alignment, conservation between mouse and human sequences was determined as the percentage of aligned nucleotide identities in mature and pre-miRNA sequences.

### RTPCR on key gene targets
Total RNA samples were prepared from the same RNA isolated for the bulk RNA-seq as described above and were DNase I treated using TRIzol according to the manufacturer's instructions. cDNA synthesis was performed using the BioRad iScript™ Reverse Transcription Supermix. To quantify gene expression of genes using TaqMan probes BioRad iTaq™ Universal Probes Supermix was used to perform quantitative real-time PCR. To quantify gene expression of genes using DNA probes BioRad iTaq™ Universal SYBR® Green Supermix was used to perform quantitative real-time PCR. ACTB, GAPDH, and 18S were used as an internal reference. Quantitative real-time PCR reactions were performed on an BioRad CFX384 Real-Time PCR Instrument. Delta-delta-cycle threshold (ΔΔCT) was determined relative to ACTB, GAPDH, and 18S levels and normalized to control samples. Error bars indicate the standard deviation of the mean from at least three biological replicates.

### Gene Set Enrichment Analysis (GSEA)
For pathway analysis, we utilized fast Gene Set Enrichment Analysis (fGSEA)[101]. Pathway analysis was done utilizing both MSigDB gene sets[46] (specifically, Hallmark and KEGG) and custom-made Gene Set files from MitoPathway and expert-curated mitochondrial genes available in[89]. Using fGSEA, all samples were compared to controls and the ranked list of genes was defined by the t-score statistics. The statistical significance was determined by 1000 permutations of the genesets[89]. We utilized a statistical cutoff of FDR < 0.05 as recommended by GSEA and lollipop plots were made using ggplot2 (ver 3.3.5). All GSEA data generated for this manuscript can be found in Supplemental Data 3.

### Murine simulated space environment experiments
The murine data utilized here were from experiments previously reported by Malkani et al.[13] and Paul et al.[14]. All experiments were approved by Brookhaven National Laboratory's (BNL) Institutional Animal Care and Use Committee (IACUC) (protocol number: 506) and all experiments were performed by trained personnel in AAALAC accredited animal facilities at BNL, while conforming to the U.S. National Institutes of Health Guide for the Care and Use of Laboratory Animals. All methods were carried out in accordance with the relevant guidelines and regulations and are reported in accordance with ARRIVE guidelines. Briefly, $n = 80$ 15-week +/−3-day old, C57Bl/6J wildtype female mice were purchased from Jackson Laboratories and housed at Brookhaven National Laboratory (BNL, Upton, NY). Upon arrival to BNL, mice were quarantined and acclimated to a standard 12:12 h light:dark cycle, with controlled temperature/humidity for

1-week prior to cage acclimation. Food and water were given *ad libitum*, and standard bedding was changed once per week. The normally loaded (NL) mice were originally utilized in parallel experiments with hindlimb unloaded mice which were not reported in this manuscript. Mice were cage acclimated ($n = 10$ mice per group; 2 mice per cage to maintain social interaction) 3-days prior to HU, followed by 14-days either normally loaded (NL) or hindlimb unloaded. Irradiation was administered on day 13 and blood tissues were collected at 24-h post-irradiation and post-euthanasia by $CO_2$ overdose, followed by cervical dislocation. Blood was collected via the abdominal aorta in EDTA-coated tubes (0.5 M) and plasma was separated by centrifugation at $2000 \times g$ for 15 min. Plasma was collected and flash frozen in liquid nitrogen for −80 °C storage. A 100 µl aliquot of cellular fraction was flash frozen and stored at −80 °C for RNA analyses, while the remaining cellular fraction was lysed with 1× RBC lysis buffer (Thermo Fisher Scientific) followed by flow cytometric preparation and analyses, as described below. All organs were flash frozen and collected at the time of dissection. All tissues were stored at −80 °C. Body weight tracking was performed on days −3, 0, 7 and 14.

On day 13, mice were transported on BNL base to the NASA Space Radiation Laboratory (NSRL) facility by animal care staff and were transferred to individual HU boxes. The following doses of irradiation were administered; GCR (0.5 Gy) and Sham control (0 Gy). Mice were positioned in the plateau region of the Bragg curve and irradiated at room temperature in individual HU boxes to enable whole body irradiation while maintaining hindlimb suspension. The NSRL physics staff performed the dosimetry. This dose of radiation is equivalent to what an astronaut is predicted to receive in deep space during a Mars mission, though it is modeled as a single exposure over 25 min instead of the actual chronic exposures over 1.5 years. A $60 \times 60$ beam was utilized at the NSRL for irradiation. Sham controls were treated similarly to GCR irradiated mice, including HU cage boxes and beam line (without irradiation) for the same duration as GCR simulation, i.e. 25 min.

### miRNA extraction from murine tissues
MiRNA extractions from plasma and all tissues were carried out using the Qiagen miRNeasy serum/plasma kit (Cat# 217184). MiRNA extractions from dissolved microvessel constructs were carried out using the Qiagen miRNeasy Mini Kit (Cat# 217004). Quantitation of miRNA samples was done using a NanoDrop 2000 Spectrophotometer (ThermoFisher Scientific).

### miRNA sequencing on murine samples
Library construction and sequencing was performed from miRNAs isolated from plasma, liver, heart, and soleus muscle from the mouse experiments described above. The miRNA extraction was carried out using the QIAgen miRNeasy kit (#217004). The total RNA quality and quantity were analyzed using a Bioanalyzer 2100 (Agilent, CA, USA) with RIN number >7. Approximately 1 µg of total RNA was used to prepare a small RNA library according to the protocol of TruSeq Small RNA Sample Prep Kits (Illumina, San Diego, USA). Single-end sequencing was performed using 50 bp on an Illumina Hiseq 2500 at the LC Sciences (Hangzhou, China) following the vendor's recommended protocol. All miRNA-sequence raw data was deposited on NASA OSDR's Biological Data Management Environment with the following identifiers: for all heart tissue related data: OSD-334, DOI: 10.26030/cg2g-as49; for all liver tissue related data: OSD-335, DOI: 10.26030/72ke-1k67: for all soleus muscle related data: OSD-337, DOI: 10.26030/m73g-2477; and for all plasma related data: OSD-336, DOI: 10.26030/qasa-rr29.

### Analysis of miRNA sequencing from murine samples
Raw reads were subjected to an in-house software program, ACGT101-miR (LC Sciences, Houston, Texas, USA) to remove

adapter dimers, junk, low complexity, common RNA families (rRNA, tRNA, snRNA, snoRNA) and repeats. Subsequently, unique sequences with length in 18-26 nucleotide were mapped to specific species precursors in miRBase 22.0[100] by BLAST search to identify known miRNAs and novel 3p- and 5p- derived miRNAs. Length variation at both 3' and 5' ends and one mismatch inside of the sequence were allowed in the alignment. The unique sequences mapping to specific species mature miRNAs in hairpin arms were identified as known miRNAs. The unique sequences mapping to the other arm of known specific species precursor hairpin opposite to the annotated mature miRNA-containing arm were considered to be novel 5p- or 3p-derived miRNA candidates. The remaining sequences were mapped to other selected species precursors (with the exclusion of specific species) in miRBase 22.0 by BLAST search, and the mapped pre-miRNAs were further BLASTed against the specific species genomes to determine their genomic locations. The above two were defined as known miRNAs. The unmapped sequences were BLASTed against the specific genomes, and the hairpin RNA structures containing sequences were predicated from the flank 80 nt sequences using RNAfold software (http://rna.tbi.univie.ac.at/cgi-bin/RNAWebSuite/RNAfold.cgi). The criteria for secondary structure prediction were: (1) number of nucleotides in one bulge in stem (≤12); (2) number of base pairs in the stem region of the predicted hairpin (≥16); (3) cutoff of free energy (kCal/mol ≤−15); (4) length of hairpin (up and down stems + terminal loop ≥50); (5) length of hairpin loop (≤20); (6) number of nucleotides in one bulge in mature region (≤8); (7) number of biased errors in one bulge in mature region (≤4); (8) number of biased bulges in mature region (≤2); (9) number of errors in mature region (≤7); (10) number of base pairs in the mature region of the predicted hairpin (≥12); and (11) percent of mature in stem (≥80). Differential expression of miRNAs based on normalized deep-sequencing counts was analyzed by selectively using Fisher exact test, Chi-squared $2 \times 2$ test, Chi-squared nXn test, Student's $t$ test, or ANOVA based on the experimental design. The significance threshold was set to be 0.01 and 0.05 in each test.

To determine gene ontology (GO)[102] and C2 pathways being regulated by the miRNAs, we performed miRNA gene set analysis utilizing the RbiomirGS[23] v0.2.12 R package from the processed miRNA analysis for all conditions in the plasma and heart tissues. The RbiomirGS package has been extensively shown in the literature to provide strong pathway prediction that provide strong biological results[103–106]. From the pathways we chose an FDR < 0.25 cutoff for significantly regulated pathways. We plotted the specific pathways with R package ggplot2.

## Weighted Gene Co-expression Analysis (WGCNA) of miRNAs-seq murine data

Weighted Gene Co-expression Analysis (WGCNA) was performed on the OSDR-334, OSDR-335, OSDR-336, and OSDR-337 miRNA-Seq datasets. First, the miRNA-Seq datasets were merged and outlier samples and genes were removed based on manual inspection using hierarchical clustering, Principal Component Analysis (PCA), and the built-in WGCNA goodSamplesGenes() function. Then Variance Stabilizing Transformation (VST) was performed using the R Bioconductor package, DESeq2 version 1.38.3. The R Bioconductor package, WGCNA version 1.72-1 was used to perform WGCNA following the standard protocol as detailed in the package vignette.

## NASA Twin Study data RNA-seq analysis

Specific details related to all the methods related to the NASA Twin Study RNA-seq data can be found in the following references[13,17]. Briefly, we will highlight the key methods. The NASA Twins Study involved two male twin subjects, aged 50 years old at the time of launch. The flight twin spent 340 days aboard the ISS while his identical twin stayed on Earth as the ground control.

Blood samples were collected into 4 mL CPT (Cell Preparation Tube with Sodium HeparinN) tubes (BD Biosciences Cat # 362760) per manufacturer's recommendations. Cell separation was performed by centrifugation in at $1800 \times g$ for 20 min at room temperature, both on the ISS and for the ground-based samples. Ambient blood collected samples slated for immediate return on Soyuz capsule were stored at 4 °C until processing (average of 35–37 h after collection, including repatriation time). Samples collected on Earth and the ISS and planned for long-term storage were mixed by inversion and immediately frozen at −80 °C.

Fresh processing of CPT tubes was performed using the following steps: 1. plasma was retrieved from the CPT tubes and flash frozen prior to long term storage at −80 °C. 2. The peripheral blood mononuclear cells (PBMCs) were recovered and washed in PBS. 0.5 million PBMCs were retrieved from one pre-flight and one post-flight samples, pelleted, flash frozen and stored at −80 °C until use for RNA extractions. The flow through from cell sorting steps was recovered as the lymphocyte depleted (LD) fraction; LD and PBMC cell specimens were lysed into RLT+ buffer (Qiagen Cat# 1053393), flash frozen and stored at −80 °C until use. Details of transcriptome preparation, sequencing, and analysis were described previously[17].

## NASA Twin Study data miRNA-seq analysis

Specific details related to all the methods related to the NASA Twin Study miRNA-seq data can be found in the following references[13,17]. Briefly, we will highlight the key methods. Small RNA libraries were prepared from 50 ng total RNA using the NEBNext Multiplex Small RNA Library Prep Set for Illumina (NEB #E7560) per manufacturer's recommendations with the following modifications: adapters and RT primer were diluted four fold, 17 cycles of PCR were used for library amplification, and no size selection was performed. The i7 primers in the NEBNext Multiplex Oligos for Illumina Dual Index Primers (NEB# E7600, NEB#E7780) were used to supplement the index primers in NEB #E7560. The libraries were sequenced in an Illumina NextSeq instrument (1x50bp).

Standard libraries were preprocessed and quality controlled using miRTrace[107]. Subsequently, reads were mapped against MirGeneDB sequences[108] using the miRDeep2[109] quantifier module. Expression values were normalized to reads per million (RPM) considering only miRNA counts. The normalized RPM value were utilized for all analysis. If the value for the miRNA was zero for all samples that miRNA was excluded from the analysis. To determine statistically significant miRNAs, ANOVA analysis with $p$-value < 0.2 was independently performed for each cell type. For flight only comparisons, the same statistical significance was applied for all time points excluding the ground samples and all ambient return samples. All statistics were run independently for each cell condition/type. To determine the overlap of the miRNA signature in this paper with the Twin Study miRNA-seq data, the proposed miRNAs were used as well as all mature components of the miRNAs included in the miRNA family.

## Inspiration4 (i4) astronaut sample collection

Detailed methods for sample collection and data processing are described in Overbey et al.[19], and Kim et al.[110]. Blood samples were collected before (Pre-launch: L-92, L-44, and L-3) and after (Return; R + 1, R + 45, and R + 82) the spaceflight. There were two male and two female astronauts for this mission. Chromium Next GEM Single Cell 5' v2, 10x Genomics was used to generate single cell data from isolated PBMCs. Subpopulations were annotated based on Azimuth human PBMC reference.

In summary, blood samples were collected before (prelaunch; L-92 days, 44 days, and 3 days) and after (return; R + 1 day, R + 45 days, and R + 82 days) the spaceflight. 21 miRNA target genes suggested in the paper has been used for identifying DEGs from i4 PBMCs and subpopulations. The Seurat method was used to normalize RNA count

data and calculate average expression of each gene. We used the average expression of the genes from four astronauts comparing post-flight (R + 1) vs pre-flight (L-92, L-44, and L-3) to identify DEGs with the Wilcoxon signed-rank test ($p < 0.05$). Average expression of the 21 miRNA target genes was used to generate heatmap plotted by the pheatmap R package. Figure 6f was created with BioRender.com.

For skin spatial transcriptomics data, 4 mm diameter skin biopsies were obtained from all Inspiration4 crew members, once before flight and as soon as possible after return (L-44 and R + 1). These biopsies were flash frozen and processed with the NanoString GeoMx platform. A 20x scan was used to select 95 freeform regions of interest (ROIs) to guide selection of outer epidermal (OE), inner epidermal (IE), outer dermal (OD) and vascular (VA) regions. GeoMx WTA sequencing reads from NovaSeq6000 were compiled into FASTQ files corresponding to each ROI and converted to digital count conversion files using the NanoString GeoMx NGS DnD Pipeline. From the Q3 normalized count matrix that accounts for factors such as capture area, cellularity, and read quality, the DESeq2 method was used to perform differential expression analysis. The procedure followed guidelines set by Health Insurance Portability and Accountability Act (HIPAA) and operated under Institutional Review Board (IRB) approved protocols and informed consent was obtained. Experiments were conducted in accordance with local regulations and with the approval of the IRB at the Weill Cornell Medicine (IRB #21-05023569).

### JAXA Cell-Free Epigenome (CFE) Study RNA quantification data

Aggregated RNA differential expression data and study protocols were shared through NASA's Open Science Data Repository with accession number: OSD-530[18]. Plasma cell-free RNA samples for RNA-seq analysis were derived from blood samples collected from 6 astronauts before, during, and after the spaceflight on the ISS. Mean expression values were obtained from normalized read counts of 6 astronauts for each time point. Heatmaps were made for the 21 genes key genes on the normalized values per time point using R package pheatmap version 1.0.12.

### Reporting summary

Further information on research design is available in the Nature Portfolio Reporting Summary linked to this article.

## Data availability

The RNA-seq data from the 3D microvessel cell model data and the JAXA CFE data are available via the NASA Open Science Data Repository's (OSDR)'s Biological Data Management Environment[94] with accession numbers: OSD-577[111], DOI: 10.26030/rs3g-e189 and OSD-530[18], https://doi.org/10.26030/r2xr-h714. The murine miRNA-seq data was also deposited on NASA OSDR's with the following identifiers, for all heart tissue related data: OSD-334[112], DOI: 10.26030/cg2g-as49; for all liver tissue related data: OSD-335[113], DOI: 10.26030/72ke-1k67; for all soleus muscle related data: OSD-337[114], DOI: 10.26030/m73g-2477; and for all plasma related data: OSD-336[115], https://doi.org/10.26030/qasa-rr29. Deposited data from the sequencing data from the NASA Twin Study can be found on the NASA Life Sciences Data Archive (LSDA) and the accession code is not available due to privacy concerns. LSDA is the repository for all human and animal research data, including that associated with this study. LSDA has a public facing portal where data requests can be initiated[116]. The LSDA team provides the appropriate processes, tools, and secure infrastructure for archival of experimental data and dissemination while complying with applicable rules, regulations, policies, and procedures governing the management and archival of sensitive data and information. The LSDA team enables data and information dissemination to the public or to authorized personnel either by providing public access to information or via an approved request process for information and data from the LSDA in accordance with NASA Human Research Program and JSC Institutional

Review Board direction. The Inspiration4 data has been uploaded to two data repositories: the NASA Open Science Data Repository (osdr.nasa.gov; comprised of NASA GeneLab and the NASA Ames Life Sciences Data Archive [ALSDA]), and the TrialX database. Identifiers for publicly downloadable datasets in the OSDR are documented as follows: 1) Data can be visualized online through the SOMA Browser (https://epigenetics.weill.cornell.edu/apps/I4_Multiome/), the single-cell browser (https://soma.weill.cornell.edu/apps/I4_Multiome/), and the microbiome browser (https://soma.weill.cornell.edu/apps/I4_Microbiome/). 2) For the PBMC data the data is available with OSDR accession ID: OSD-570 and the following link: https://osdr.nasa.gov/bio/repo/data/studies/OSD-570/.

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

## Acknowledgements

We thank Diogo de Moraes for curating the senescence cell markers. This work was supported by NASA grant 16-ROSBFP_GL-0005: NNH16ZTT001N-FG Appendix G: Solicitation of Proposals for Flight and Ground Space Biology Research (Award Number: 80NSSC19K0883) and The Translational Research Institute for Space Health through NASA Cooperative Agreement NNX16AO69A (T-0404) awarded to A.B. Orphan gene analysis was supported by National Science Foundation IOS 1546858 awarded to E.S.W. M.A.M. was funded by Conselho Nacional de Desenvolvimento Científico e Tecnológico (CNPq) (310287/2018-9) and Fundação de Amparo à Pesquisa do Estado de São Paulo (FAPESP) (2021/08354-2). The opinions expressed in this article are those of the authors and do not reflect the view of the National Institutes of Health, the Department of Health and Human Services, or the United States government. JK was supported by Basic Science Research Program through the National Research Foundation of Korea(NRF) funded by the Ministry of Education(RS-2023-00241586). CEM thanks the WorldQuant Foundation, NASA (NNX14AH50G, NNX17AB26G, 80NSSC22K0254, NNH18ZTT001N-FG2), the National Institutes of Health (R01MH117406, P01CA214274, R01CA249054), and the LLS (LLS SCOR-7027-23, LLS SCOR 7029-23). JK acknowledge Boryung for their financial support and research enhancement ground, provided through their Global Space Healthcare Initiative, Humans In Space, including mentorship and access to relevant expert networks. S.B.W. and K.S. acknowledges this work was partially funded by the UK Space Agency through a grant [ST/X000036/1] administered by the Science and Technology Facilities Council (STFC). S.B.W. is supported by Kidney Research UK [RP_017_20190306; ST_001_20221128; TF_007_20191202]. K.S. acknowledges this research was funded in part by the Wellcome Trust [Grant number 110282/Z/15/Z]. For the purpose of open access, the author has applied a CC BY public copyright license to any Author Accepted Manuscript version arising from this submission.

## Author contributions

Conceptualization: A.B. and P.G.; Methodology: A.B., and P.G.; Formal Analysis: A.B., J.T.M., J.K., J.H., F.J.E., V.Z., N.S.T., S.A., D.G., R.E.S., S.T., Y.B., C.M., K.R., K.S., and J.P.; Investigation: A.B., C.E.M., P.G., M.M., F.J.E., V.Z., E.S.W., J.K., J.P., D.G., R.E.S., S.T., Y.B., and J.T.M.; I4 and NASA Twin Study data and omics: C.E.M., C.M., E.O., J.P., and J.K.; JAXA data: M.M.; Resource: A.B., R.E.S., C.E.M., P.G., and M.M.; Original Draft: J.T.M., P.G., L.F., M.L.J., and A.B.; Review & Editing: J.T.M, J.K, L.F., M.L.J., N.S.T., S.A., K.S., S.T., Y.B., J. P., E.O., J.H., U.S, F.J.E., V.Z., J.W.G, M.T., D.C.W., C.M., S.B., R.M., M.M., D.M.P., B.K., M.A.M., S.B.W., D.S.R., S.M., M.B., C.A.M., E.S.W., R.E.S., D.G., C.E.M., P.G., A.B.; Figures and Visualization: A.B., J.T.M., J.K., F.J.E., N.S.T., D.G., S.A., R.E.S., and J.P.; Funding Acquisition: A.B., C.E.M. (for i4 study), and P.G.; Supervision: A.B.

## Competing interests

The authors declare no competing interests.

## Additional information

¹Department of Radiation Medicine, Georgetown University School of Medicine, Washington, D.C, USA. ²Department of Physiology, Biophysics and Systems Biology and the WorldQuant Initiative, Weill Cornell Medicine, New York, NY, USA. ³Vascular Medicine Institute at the University of Pittsburgh Department of Medicine, Pittsburgh, PA, USA. ⁴Department of Bioengineering, University of Pittsburgh, Pittsburgh, PA 15213, USA. ⁵Division of International Epidemiology and Population Studies, Fogarty International Center, National Institutes of Health, Bethesda, Maryland, USA. ⁶Center for Data-Driven Discovery in Biomedicine, Children's Hospital of Philadelphia, Philadelphia, PA, USA. ⁷Division of Neurosurgery, Children's Hospital of Philadelphia, Philadelphia, PA, USA. ⁸London Tubular Centre, Department of Renal Medicine, University College London, London, UK. ⁹Division of Gastroenterology and Hepatology, Department of Medicine, Weill Cornell Medicine, New York, NY, USA. ¹⁰The Center for Mitochondrial and Epigenomic Medicine, The Children's Hospital of Philadelphia,

Philadelphia, PA 19104, USA. [11]Bioinformatics and Computational Biology Program, Department of Genetics, Development and Cell Biology, Iowa State University, Ames, IA 90011, USA. [12]Instituto de Medicina Molecular João Lobo Antunes, Faculdade de Medicina, Universidade de Lisboa, 1649-028 Lisboa, Portugal. [13]Center for Translational Data Science, University of Chicago, Chicago, IL 60637, USA. [14]Clever Research Lab, Springfield, IL 62704, USA. [15]Departments of Oncology and Medicine and the Sidney Kimmel Comprehensive Cancer Center, The Johns Hopkins Medical Institutions, Baltimore, MD, USA. [16]Department of Pediatrics, Division of Human Genetics, University of Pennsylvania School of Medicine, Philadelphia, PA 19104, USA. [17]Neuroscience Institute, Department of Neurobiology/ Department of Pharmacology and Toxicology, Morehouse School of Medicine, Atlanta, GA 30310, USA. [18]Transborder Medical Research Center, University of Tsukuba, Ibaraki 305-8575, Japan. [19]Department of Genome Biology, Institute of Medicine, University of Tsukuba, Ibaraki 305-8575, Japan. [20]Department of Agricultural and Biological Engineering, Purdue University, West Lafayette, IN 47907, USA. [21]Department of Biochemistry and Tissue Biology, Institute of Biology, Universidade Estadual de Campinas, Campinas, SP, Brazil. [22]Obesity and Comorbidities Research Center (OCRC), Universidade Estadual de Campinas, Campinas, SP, Brazil. [23]LBTI- UMR CNRS 5305, Université Claude Bernard Lyon 1, Lyon, France. [24]ICBMS, UMR5246, CNRS, INSA, CPE-Lyon, Université Claude Bernard Lyon 1, Villeurbanne, France. [25]Department of Experimental Medicine, University of Rome Tor Vergata, 00133 Rome, Italy. [26]3d.FAB, CNRS, INSA, CPE-Lyon, UMR5246, ICBMS, Université Claude Bernard Lyon 1, Villeurbanne, France. [27]Genetics Program, Department of Genetics, Development and Cell Biology, Iowa State University, Ames, IA 90011, USA. [28]Facultad de Ingeniería, Universidad Nacional de Asunción, San Lorenzo, Paraguay. [29]Center for Radiological Research, College of Physicians and Surgeons, Columbia University, New York, NY, USA. [30]Stanley Center for Psychiatric Research, Broad Institute of MIT and Harvard, Cambridge, MA, USA. [31]Blue Marble Space Institute of Science, Space Biosciences Division, NASA Ames Research Center, Moffett Field, CA, US. [32]These authors contributed equally: J. Tyson McDonald, JangKeun Kim, Lily Farmerie, Meghan L. Johnson. [33]These authors jointly supervised this work: Peter Grabham, Afshin Beheshti. ✉e-mail: pwg2@cumc.columbia.edu; afshin.beheshti@bmsis.org

