## [Peer Review File · Nature Communications]

Space Radiation Damage Rescued by Inhibition of Key Spaceflight Associated miRNAsREVIEWER COMMENTS

Reviewer #1 (Remarks to the Author):

In this manuscript, authors investigated the role of three miRNAs in space radiation-induced damage in astronauts. Though the study is correlative, the results presented in this manuscript are exciting. The most important aspect of this study is the availability of samples and validation of results in astronauts. Authors should consider doing in-depth mechanistic experiments before the manuscript can be considered for publication in Nature Communications.

1. Authors should consider validating at least some of the predicted targets by qPCR, western blot, and luciferase assay. Without validating a few of the targets its a giant leap of faith as target identification is the Achilles Heel of the miRNA biology.

2. Authors should consider quantifying the number of 53BP1 foci. They should also consider showing the co-localization of γ -H2AX and 53BP1.

3. To further substantiate their claims of NER DNA repair pathways being the main denominator of GCR-induced changes in astronauts, authors should consider doing functional GFP-based assays for NER and compare with another DNA repair pathway like NHEJ, which is mostly involved in repairing radiation-induced damaged DNA.

listing

4. Fig. 3E-N: Authors should consider listing the genes in the same order for all panels. it will be helpful for readers to compare the effects on different immune cells.

Reviewer #2 (Remarks to the Author):

McDonald et al evaluated the potential utility of three antagomiRs as mitigators of space radiation induced DNA damage and genotoxic stress. The data presented include changes in gene expression after 0.5 Gy GCR exposure in primary HUVEC cells with and without transfection with three antagomirs, gene expression analysis of multiple tissues collected from irradiated mice, and analysis of specimens collected from astronauts during or after space flight. A large amount of data (mostly discordant) are presented. It is not clear what the authors are trying to communicate? Are they saying that a single administration of three antagomirs will mitigate the toxicity from space radiation? If so, that is difficult to conceive, considering its effects on organism as a whole. There are major flaws in experimental design, data analysis and even more in data interpretation (OR over interpretation). Major concerns:

1. Radiation induced DNA breaks are repaired mostly in 30 min (including many complex damages) and time points used for gene expression analysis does not fit with repair kinetics.

2. Experiments are not well controlled, and the data is cursory. What is the baseline expression of miRNAs evaluated and how much antagomirs got into? The changes in critical targets following the antagomir transfection need to be shown at protein level (Western blots). One miRNA can target several genes and several genes can be targeted by one miRNA.

3. miRs tested are known to be altered in many diseases (e.g., miR16- leukemia, miR-125- vascular diseases, let-family in multiple disease). Dysregulation of these using antagomirs will result in undesired complications at the cellular and organism level and not "to thank for all the antagomirs".

4. What is the connection between the data from in vitro studies, mouse tissue and those generated in astronauts' specimens? Investigator team has access to astronaut's specimens, which is a plus. If there are variations (if any), are they resulting from radiation induced damage. Are these astronauts getting an acute 0.5 Gy GCR exposure?

Minor comments:

1. Figures and data presented in a haphazard manner. Fonts used in figures (except Figure 5) is not legible

Reviewer #3 (Remarks to the Author):

McDonald et al.

Summary

In their recent papers, the Beheshti group has shown interesting miRNA signatures associated with spaceflight. They previously showed that three antagomirs against miR-125, miR-16, and let-7a could reduce cardiovascular damage induced by radiation in the mature cell model (Malkani et al., Cell Reports, 2020; Reference 12 in this manuscript) and the angiogenesis model (Wuu et al., iScience, 2020; Reference 14 in this manuscript) using the 3D HUVEC vessel system. Here, they proposed to determine critical biological factors rescued by the antagomir treatment. The authors performed and analyzed several miRNA-seq and RNA-seq using mouse and HUVEC cell models. However, their analyses provided only indirect evidence or predicted effects rather than clear molecular targets of the antagomirs in this manuscript. Therefore, I do not support its publication in Nature Communications.

Major points

1. One of the new data in this manuscript is that individual antagomirs also could mitigate the structural damage caused by the radiation (Fig 1A and B). Therefore, it is expected to do the 53BP1 foci experiment (Fig 1E) using not only the three antagomir mixture but also individual antagomirs. It will provide additional insight into how three miRNAs differentially (or similarly) affect the process.
2. The authors performed miRNA-seq using mouse tissues and identified differentially expressed miRNAs upon irradiation. To determine pathways regulated by those miRNAs, the authors performed miRNA gene set analysis using RBiomirGS (Reference 72). As far as I understand, RBiomirGS receives miRNA fold-changes and p-values and then returns predicted target genes and their gene set enrichments. This approach may be helpful in predicting the possible effects of miRNAs but is not appropriate for understanding precise molecular mechanisms in a specific cell context because the expression level of each miRNA and target RNA was not considered in this analysis. The proper way is to perform simultaneous miRNA-seq and RNA-seq using the normal and irradiated mouse tissues and to analyze the differentially expressed mRNAs which contain the seed sequences (8mer, 7mer-A1, and 7mer-m8 sites; see Bartel, Cell, 2018, PMID: 29570994) of differentially expressed miRNAs.
3. The authors have used FDR < 0.25 in many figures, including Fig 1F, 1G, 4A, 4B, 5A, 6A, and so on. I am not sure whether they can be considered significant results.
4. I think the volcano plot in Fig 2D is the most critical figure in this manuscript. There are no significantly changed genes between 0.5 Gy vs 0 Gy in the angiogenesis model. Also, only a few genes are changed between 0.5 Gy vs 0 Gy in the mature model. It implies that irradiation does not dramatically change the transcriptome. (1) I wonder how PCA in Fig 2B shows different patterns between 0.5 Gy and 0 Gy. (2) Fig 4, 5, and 6 depend on the same RNA-seq, and I wonder how we could see enriched gene sets between 0.5 Gy and 0 Gy in the angiogenesis model (For example, Fig 4D). (3) I wonder whether it is appropriate to describe the changes as 'restoration' because there were no changes after irradiation. Because of this issue, I doubt whether the changes in Fig 4, 5, and 6 are physiologically meaningful.
5. Scatter plots (TPM; Transcripts Per Million) are required to show the RNA-seq pattern in Fig 2D clearly. Label 21 genes on the plot.

6. Bar graphs are needed to show the actual expression levels (TPM) upon the treatment of irradiation and antagomirs in Fig 2F.
7. Validation experiments are required using another method, such as qRT-PCR in Fig 2F.
8. Rescue experiments are required to validate the miRNA targets. Overexpress 21 genes individually (Fig 2F) and do the same experiments in Fig 1C and 1E.
9. Because there are no target validation experiments in the current manuscript, I doubt whether it is meaningful to focus on 21 genes in Fig 3.
10. Line 229. There is no clue that miRNA was the major reason for the downregulation of the 21 genes. Why was the transcriptional regulation excluded? To answer this question, the authors are expected to analyze the RNA-seq using a cumulative plot, as shown in Fig 2C in Baek et al., Nature, 2008 (PMID: 18668037). Briefly, assign mRNA groups according to miRNA seed sequences, draw a cumulative plot, and do the Kolmogorov–Smirnov test. This analysis will determine whether miRNA is the most crucial regulator in this context.
11. The authors used a t-score in the heatmap in Fig 5B but didn't label it on the figure. Usually, people expect a log fold-change in this kind of heatmap. Please draw heatmaps using fold changes.
12. The authors are expected to provide supplementary tables containing raw and processed data.

Minor points

1. Fig. 1D. Describe how the authors generated this plot in the methods and the legend. What is the meaning of color? Because the font size is too small, it is not clear which genes are the common direct targets of the three miRNAs. If there are 6mer targets, they should be removed.
2. Fig 2B. The label 'Samples' is wrong.
3. Line 95. Please briefly mention the target selection criteria used in mirDIP in the text.
4. Move Fig 2G to supplementary figures.
5. Fig 2G and Fig 5B, C. If they are p-values, use adjusted p-values.
6. Expanded Fig 2. Scatter plots will be helpful in understanding the expression level of orphan genes. Pfam analysis also will be interesting.
7. Line 218. 1F -> 2F
8. Fig 3B is not mentioned in the text.
9. Line 240. 3E-O -> 3E-N. Also, these figures are appropriate in the supplementary figures.
10. Fig 4. Explain the abbreviation of NES in the legend.
11. Line 329. I don't see an increase in immune gene sets in the angiogenesis model in Fig 5A.
12. Fig 2F. Indicate the seed target types (8mer, 7mer-A1, and 7mer-m8 sites) on 21 genes.

Dear Reviewers,

I have provided below my responses to the edits requested which we have provided now with our newly submitted manuscript. My responses are in red font below the original comments. I believe the comments provided by the reviewers have strengthened the manuscript significantly.

Afshin Beheshti, PhD

REVIEWER COMMENTS

Reviewer #1 (Remarks to the Author):

In this manuscript, authors investigated the role of three miRNAs in space radiation-induced damage in astronauts. Though the study is correlative, the results presented in this manuscript are exciting. The most important aspect of this study is the availability of samples and validation of results in astronauts. Authors should consider doing in-depth mechanistic experiments before the manuscript can be considered for publication in Nature Communications.

We thank the reviewer for their comments and suggestions. We have provided additional validation and analysis (see revised Figures 3, 5, 6, 7 and Extended Figure 7), and in the future we will be conducting further mechanistic studies. Unfortunately, to perform new experiments and provide additional results with GCR irradiation, we will have to apply for beamtime at Brookhaven National Laboratory (BNL) and then conduct these experiments, which would be a very time-consuming endeavor. In addition, we will have to wait until additional funding is available to conduct more experiments. These experiments are very expensive to conduct and they can only be performed twice a year when the beamtime is available at BNL.

1. Authors should consider validating at least some of the predicted targets by qPCR, western blot, and luciferase assay. Without validating a few of the targets its a giant leap of faith as target identification is the Achilles Heel of the miRNA biology.

We thank the reviewer for this comment. We have now validated the 8 out of the 21 key genes with qPCR. All the genes except for one were confirmed to behave the same way as the RNA-seq data. Interestingly, *MSH5* showed an increase in expression for 0.5Gy compared to 0Gy, but came back to the level of 0Gy with the antagomir treatment. This indicates that although the gene was increased (which would be a false positive for this gene) the antagomir treatment still restored the levels back to control levels. As stated above, due the nature of these experiments we unfortunately cannot repeat the experiments and produce protein for western blots and luciferase assays.

2. Authors should consider quantifying the number of 53BP1 foci. They should also consider showing the co-localization of γ -H2AX and 53BP1.

We thank the reviewer for this comment. We take the opportunity to clarify that we have quantified 53BP1 foci for our experiments; the results are shown in Figure 2b. We have provided both the box plot of the data and a representative microscopy image for the 53BP1 staining. We did not stain for γ -H2AX, since typically this stain does not produce as clean of staining and results. In addition, in the radiation biology community/literature, it has been widely demonstrated that γ -H2AX and 53BP1 produce the same results when quantifying DNA DSB foci (see the following references: Penninckx S, Pariset E, Cekanaviciute E, Costes SV. Quantification of radiation-induced DNA double strand break repair foci to evaluate and predict biological responses to ionizing radiation. *NAR Cancer*. 2021 Dec 22;3(4):zcab046. doi: 10.1093/narcan/zcab046. PMID: 35692378; PMCID: PMC8693576.; E. Marková, N. Schultz & Dr I. Y. Belyaev (2007) Kinetics and dose-response of residual 53BP1/ γ -H2AX foci: Co-localization, relationship with DSB repair and clonogenic survival, *International Journal of Radiation Biology*, 83:5, 319-329, DOI: [10.1080/09553000601170469](https://doi.org/10.1080/09553000601170469); Aroumougame Asaithamby, Naoya Uematsu, Alope Chatterjee, Michael D. Story, Sandeep Burma, David J. Chen; Repair of HZE-Particle-Induced DNA Double-Strand Breaks in Normal Human Fibroblasts. *Radiat Res* 1 April 2008; 169 (4): 437–446. doi: <https://doi.org/10.1667/RR1165.1>)

Due to the nature of these experiments, it will be unfeasible to repeat them in a timely manner. To repeat these experiments with GCR irradiation, we would have to apply for beam time at Brookhaven National Laboratory and wait until additional funding is available to conduct these experiments. Since we have shown that these results are extremely reproducible, and staining for 53BP1 for DNA DSBs is a widely accepted assay, we hope for this paper that the quantification we have provided for 53BP1 will suffice.

3. To further substantiate their claims of NER DNA repair pathways being the main denominator of GCR-induced changes in astronauts, authors should consider doing functional GFP-based assays for NER and compare with another DNA repair pathway like NHEJ, which is mostly involved in repairing radiation-induced damaged DNA.

We agree with the reviewer, and conducting an in depth examination to interrogate these distinct pathways will be a future goal. However, we did not intend to infer that the alterations to NER observed in Figure 8c is the main denominator responsible for the antagomir changes to DSB observed in Figure 2b. We have revised the manuscript to clarify this, and we believe the stated interpretations for the GSEA pathway data related to changes in DNA repair are sufficiently presented without being overemphasized. Unfortunately, it is not feasible to conduct further experiments in a timely manner in order to resolve the direct impact of antagomir treatment on DNA repair pathways. To repeat these experiments and provide additional results with GCR irradiation would require additional beam time at Brookhaven National Laboratory and additional funding.

4. Fig. 3E-N: Authors should consider listing the genes in the same order for all panels. it will be helpful for readers to compare the effects on different immune cells.

We thank the reviewer for their suggestion. We now provide heatmaps with the genes in alphabetical order (figure 6i-r). The figures have been updated and revised due to reviewers comments, so this figure can now be seen in figure 6. We also provide the original clustered version of the heatmaps as a supplemental figure (Extended Figure 6). We believe that this clustered version is valuable for the reader to be able to identify which genes are closely related based on the data.

Reviewer #2 (Remarks to the Author):

McDonald et al evaluated the potential utility of three antagomiRs as mitigators of space radiation induced DNA damage and genotoxic stress. The data presented include changes in gene expression after 0.5 Gy GCR exposure in primary HUVEC cells with and without transfection with three antagomirs, gene expression analysis of multiple tissues collected from irradiated mice, and analysis of specimens collected from astronauts during or after space flight. A large amount of data (mostly discordant) are presented. It is not clear what the authors are trying to communicate? Are they saying that a single administration of three antagomirs will mitigate the toxicity from space radiation? If so, that is difficult to conceive, considering its effects on organism as a whole. There are major flaws in experimental design, data analysis and even more in data interpretation (OR over interpretation).

We thank the reviewer for their comment. We agree that the single antagomir section was not clear enough in the original manuscript, and have further addressed this in the text. This is, of course, only a small part of the story. Our goal is to shed light on how the single antagomirs perform compared to the combination of all three. Although the single antagomir treatment did have a beneficial impact, the combination of the three antagomirs had the greatest mitigation response; this is the focus of the entire paper. Based on the reviewer's suggestion, we have provided the following clarification in the text to describe this on page 4:

“Thus, in both models, miR-16-5p, miR-125b-5p, and let-7a-5p are equally effective at counteracting the effects of space radiation, but the antagomir combination inhibition of the three miRNAs provides a stronger and more robust response. Therefore, hereafter, we focus on the study of the biological changes occurring when inhibiting all three miRNAs using the combined inhibition.”

We have also added the following point to the discussion on page 14 discussing the common shared gene targets between the 3 miRNAs and the impact that has in the biology:

“There are 5 genes that are affected in common with all 3 antagomirs. When inhibited, all products of these genes are candidates for mediating the resistance to radiation damage, although the effects of other genes affected by each individual antagomir cannot be ruled out. Each gene has functions that

could contribute to GCR resistance. APAF1 and NIBAN1 are involved in the control of apoptosis and may prevent progression towards senescence. XPO6 and SLC7A1 are involved in transport across membranes between cell compartments, a process involved in early responses to cell stressors. PDPR regulates pyruvate phosphorylation and may facilitate protective functions that require a higher energy output. Overall it appears that several pathways are involved in protection and are acting in concert to push the cells into a more healthy and active biological state. “

We have also revised the paper and figures (Figs. 1 - 10) to provide a better flow of the data and have provided additional analysis to address the reviewer’s questions.

Major concerns:

1. Radiation induced DNA breaks are repaired mostly in 30 min (including many complex damages) and time points used for gene expression analysis does not fit with repair kinetics.

It is true that the peak DNA repair foci occurs at around 30 minutes after gamma irradiation. However, in this tissue model, foci induced by charged particles decline over several hours (Grabham P.W, Bigelow A., and Geard C. (2012) Effects of ionizing radiation on DNA repair dynamics in 3-Dimensional human vessel models: Differential effects according to radiation quality. *Int. J Rad Biol* 6 493 - 500. PubMed PMID 22449005). Thus, we believe that the sample time is sufficient to determine antagomir effects on DNA repair. We have added the following sentence in the results to clarify this point (Page 5, second paragraph):

“Although it is generally acknowledged that the peak DNA repair foci occur around 30 minutes after gamma irradiation, our previous publications showed that foci induced by charged particles decline over several hours in this model (REF).”

2. Experiments are not well controlled, and the data is cursory. What is the baseline expression of miRNAs evaluated and how much antigomirs got into? The changes in critical targets following the antagomir transfection need to be shown at protein level (Western blots). One miRNA can target several genes and several genes can be targeted by one miRNA.

The baseline expression of miRNAs was originally presented in our first publication on these miRNAs, which is referenced in the paper (i.e. Malkani et al, *Cell Reports*, 2021). Unfortunately due to the nature of these experiments it is not possible for this manuscript to redo experiments to get proteins from western blotting. This would require us to go to Brookhaven National Laboratory (BNL) at NASA Space Radiation Laboratory (NSRL) (as described in the methods) to conduct the irradiation experiments. To get extra beam time we will have to apply 6 months in advance for beam time at NSRL and also to obtain the budget for paying for the expensive beam time experiments. That said, we are planning to conduct more experiments with new funding in the Fall of 2024, which we are planning to publish in future manuscripts. But we believe based on our previous publications and what we are showing here,

the experiments and controls were sufficient. We have also provided additional figures throughout in response to other reviewers' comments to further expand on the impact of this work.

3. miRs tested are known to be altered in many diseases (e.g., miR16- leukemia, miR-125- vascular diseases, let-family in multiple disease). Dysregulation of these using antigomirs will result in undesired complications at the cellular and organism level and not "to thank for all the antagomirs".

We agree that dysregulation *in vivo* could involve complications due to the variety of processes the miRNAs are involved in, especially cancer, as mentioned in the manuscript (Pages 13 and 14). At this stage of the research, we are not advocating for the immediate use of antagomirs as a radioprotectant. The purpose of their use is as a mechanistic tool to unravel the pathways involved in this unique amelioration of the effects of space radiation. We have changed the title and removed "So Long, and Thanks for All the Antagomirs:".

4. What is the connection between the data from in vitro studies, mouse tissue and those generated in astronauts' specimens? Investigator team has access to astronaut's specimens, which is a plus. If there are variations (if any), are they resulting from radiation induced damage. Are these astronauts getting an acute 0.5 Gy GCR exposure?

The astronauts in this study did not receive an acute GCR dose of 0.5 Gy. The 0.5 Gy dose is comparable to the total estimated minimal dose to be received from a round-trip mission to Mars which will be influenced by solar activity and shielding (Zeitlin et al, Science 2013). We have added the following sentences in page 8 end of first paragraph describing the doses that the astronauts received for the different missions:

"It is important to note that the estimated accumulated doses of space radiation that the astronauts were exposed to for these missions were lower than 0.5Gy (which is the estimated dose for a trip to Mars and back), but despite this we still observed key changes as we will describe below. The following are the accumulated doses that were measured for each mission: NASA Twin Study astronaut received 146.34 mSv or 14.634 cGy¹⁷, I4 astronauts received 4.72 mSv or 0.472 cGy¹⁹, and for the JAXA astronauts we estimated that they had received an accumulated dose of 51.60 mSv or 5.16 cGy (the dosimetry was not reported for this mission and this is an estimate based on the average dose received per month from other missions)¹⁸."

It is not currently possible to experimentally approximate the total dose and dose-rate received during long-duration spaceflights, which routinely last several months. The ground-based experimental setup used for irradiating with GCRs currently represents the best approximation for validation of predictive models and while enabling countermeasure developments. Thus, extrapolation and comparison from the astronaut data to cell culture studies conducted on Earth as well as mice experiments both on Earth and during spaceflight are key to meaningful interpretation of biological results applicable to space flight. Lastly, it is possible that the differences in the radiation delivery might account for any deviations

between the results we are observing. To address this last point we have added a “Limitation of the Study” section at the end of manuscript discussing this point as the following:

“Limitations of the Study

Due to the nature of performing experiments in the space environment, it can be limiting and difficult to simulate perfectly the conditions in space in simulated ground experiments. So to perfectly recreate the space radiation dose and dose-rate that astronauts will receive during long-duration spaceflights (lasting several months) is not possible. The ground-based experimental setup reported in this manuscript used for irradiating with GCRs currently represents the best approximation for validation of predictive models and while enabling countermeasure developments. Thus, extrapolation and comparison from the astronaut data to cell culture studies conducted on Earth as well as mice experiments both on Earth and during spaceflight are key to meaningful interpretation of biological results applicable to space flight. Lastly, it is possible that the differences in the radiation delivery might account for any deviations between the results we are observing.”

Minor comments:

1. Figures and data presented in a haphazard manner. Fonts used in figures (except Figure 5) is not legible

We thank the reviewer for pointing this out. We have updated and revised the figures to provide a more uniform appearance and legibility. We also have added additional figures to address other reviewer’s comments and in the process have updated the majority of the figures.

Reviewer #3 (Remarks to the Author):

McDonald et al.

Summary

In their recent papers, the Beheshti group has shown interesting miRNA signatures associated with spaceflight. They previously showed that three antagomirs against miR-125, miR-16, and let-7a could reduce cardiovascular damage induced by radiation in the mature cell model (Malkani et al., Cell Reports, 2020; Reference 12 in this manuscript) and the angiogenesis model (Wuu et al., iScience, 2020; Reference 14 in this manuscript) using the 3D HUVEC vessel system. Here, they proposed to determine critical biological factors rescued by the antagomir treatment. The authors performed and analyzed several miRNA-seq and RNA-seq using mouse and HUVEC cell models. However, their analyses provided only indirect evidence or predicted effects rather than clear molecular targets of the antagomirs in this manuscript. Therefore, I do not support its publication in Nature Communications.

We thank the reviewer for their comments. We now provide major revisions to the manuscript and additional justification in support of our findings.

Major points

1. One of the new data in this manuscript is that individual antagomirs also could mitigate the structural damage caused by the radiation (Fig 1A and B). Therefore, it is expected to do the 53BP1 foci experiment (Fig 1E) using not only the three antagomir mixture but also individual antagomirs. It will provide additional insight into how three miRNAs differentially (or similarly) affect the process.

We agree with the reviewer that this is the logical next step and testing the ability of the individual antagomirs to reduce DNA damage is our next planned experiment. We also plan to test other endpoints of space radiation damage, and explore the mechanism of such effects. Unfortunately, access to such radiation at the only facility that can supply it is limited. To perform new experiments and provide additional results with GCR irradiation, we would have to apply for beamtime at Brookhaven National Laboratory (BNL), which is only available twice a year, and wait until additional funding is available to conduct these very expensive experiments.

2. The authors performed miRNA-seq using mouse tissues and identified differentially expressed miRNAs upon irradiation. To determine pathways regulated by those miRNAs, the authors performed miRNA gene set analysis using RBiomirGS (Reference 72). As far as I understand, RBiomirGS receives miRNA fold-changes and p-values and then returns predicted target genes and their gene set enrichments. This approach may be helpful in predicting the possible effects of miRNAs but is not appropriate for understanding precise molecular mechanisms in a specific cell context because the expression level of each miRNA and target RNA was not considered in this analysis. The proper way is to perform simultaneous miRNA-seq and RNA-seq using the normal and irradiated mouse tissues and to analyze the differentially expressed mRNAs which contain the seed sequences (8mer, 7mer-A1, and 7mer-m8 sites; see Bartel, Cell, 2018, PMID: 29570994) of differentially expressed miRNAs.

We thank the reviewer for this suggestion. We now provide the specific analysis on the RNA-seq with the miRNA gene targets as requested. This is added to the new Figure 5. The addition of this analysis has also provided a way to validate that, with the antagomir treatment, the 21 target genes are no longer targeted by the 3 miRNAs. Due limited budget we did not perform miRNA-seq on the 3D tissues and this will be planned for future experiments.

In addition, we have also provided additional data from the NASA Twin Study which had both RNA-seq and miR-seq performed on different components from the blood. We have done additional analysis to correlate the three miRNAs and the 21 genes and see results that would be expected if the miRNAs were targeting the 21 genes (Fig. 7 and Extended Data Figure 7). We have added the following paragraph describing the results in the results section discussing the human data:

“To establish the direct relationship between the 21 gene targets and the three miRNAs we utilized the NASA Twin Study which had both miRNA- and mRNA-seq data associated with the mission and cell types. Similar to the cumulative plots, we observed the correlation between the three miRNAs (i.e. miR-16, let-7a, and miR-125b) with the 21 gene targets in the four different cell types (i.e. CD4, CD8, CD19, and monocytes) (Extended Data Fig. 7b). We correlated with the overlapping time points per each sample type and observed for the majority of the miRNAs and genes targets an inverse relationship. Specifically, for monocytes for the specific gene targets the miRNAs had an inverse correlation (Fig. 7b). Since the miRNAs will traditionally silence the specific gene target this result is what we would expect and provides further validation in astronauts that these miRNAs in circulation will be ideal targets to reduce health risks as our in vitro results show.”

3. The authors have used $FDR < 0.25$ in many figures, including Fig 1F, 1G, 4A, 4B, 5A, 6A, and so on. I am not sure whether they can be considered significant results.

To accommodate the reviewer’s comment, we have updated the plots to show $FDR < 0.05$ and have provided all the Gene Set Enrichment Analysis (GSEA) values as supplemental data for readers to reference. That said, we would like to bring to the attention of the reviewer that showing $FDR < 0.25$ in fact is the correct statistics showing GSEA results. The plots that the reviewer is referring to represent pathway analysis through GSEA. This method is widely accepted in the community. The scientists that created the GSEA algorithm consider an appropriate cutoff for statistics to be $FDR < 0.25$.

The GSEA FAQ question that specifically addresses this point can be found here:

https://software.broadinstitute.org/cancer/software/gsea/wiki/index.php/FAQ#Why_does_GSEA_use_a_false_discovery_rate_.28FDR.29_of_0.25_rather_than_the_more_classic_0.05.3F. They specifically state the following:

“Why does GSEA use a false discovery rate (FDR) of 0.25 rather than the more classic 0.05?”

An FDR of 25% indicates that the result is likely to be valid 3 out of 4 times, which is reasonable in the setting of exploratory discovery where one is interested in finding candidate hypothesis to be further validated as a results of future research. Given the lack of coherence in most expression datasets and the relatively small number of gene sets being analyzed, using a more stringent FDR cutoff may lead you to overlook potentially significant results. For more information about gene set enrichment analysis results, see Interpreting GSEA in the GSEA User Guide.”

A further explanation is given in the GSEA user guide which is found here: https://www.gsea-msigdb.org/gsea/doc/GSEAUUserGuideFrame.html?Interpreting_GSEA.

We have provide below what is stated specifically on the FDR:

“The false discovery rate (FDR) is the estimated probability that a gene set with a given NES represents a false positive finding. For example, an FDR of 25% indicates that the result is likely to be valid 3 out of 4 times. The GSEA analysis report highlights enrichment gene sets with an FDR of less than 25% as those most likely to generate interesting hypotheses and drive further research, but provides analysis results for all analyzed gene sets. In general, given the lack of coherence in most expression datasets and the relatively small number of gene sets being analyzed, an FDR cutoff of 25% is appropriate. However, if

you have a small number of samples and use gene_set permutation (rather than phenotype permutation) for your analysis, you are using a less stringent assessment of significance and would then want to use a more stringent FDR cutoff, such as 5%.”

4. I think the volcano plot in Fig 2D is the most critical figure in this manuscript. There are no significantly changed genes between 0.5 Gy vs 0 Gy in the angiogenesis model. Also, only a few genes are changed between 0.5 Gy vs 0 Gy in the mature model. It implies that irradiation does not dramatically change the transcriptome. (1) I wonder how PCA in Fig 2B shows different patterns between 0.5 Gy and 0 Gy. (2) Fig 4, 5, and 6 depend on the same RNA-seq, and I wonder how we could see enriched gene sets between 0.5 Gy and 0 Gy in the angiogenesis model (For example, Fig 4D). (3) I wonder whether it is appropriate to describe the changes as ‘restoration’ because there were no changes after irradiation. Because of this issue, I doubt whether the changes in Fig 4, 5, and 6 are physiologically meaningful.

This assessment isn’t exactly correct. Firstly, the radiation treatment was given six days before the endpoint in the angiogenesis model whereas only 24 hours have elapsed for the mature model after radiation. In fact, there are a number of genes with a p-value < 0.05 and FC >1.2 in both models. As the reviewer may know, the adjusted p-values are put in place for omics as a stringent method to reduce false positives with omics data. With the adj. p-value < 0.05 there were no significant genes as the reviewer commented. But that does not mean no biological changes are occurring. Sometimes due to experimental variation, when applying more stringent statistics the biology can be lost as is the case here. When looking at the pathway analysis with Gene Set Enrichment Analysis (GSEA) (as described above), we see many pathways being regulated for the 0.5Gy vs 0 Gy angiogenesis comparison. This is because the GSEA analysis determines significantly enriched pathways with information on how all the genes (significant or not) are driving that pathway. Meaning more subtle changes can have an impact on biology. So indeed the changes observed are very meaningful which most bioinformaticians will agree with due to how the individual genes will impact overall pathways.

For this reason we have decided to lower the statistical significance for figure 4 to p-value for these plots. As indicated in the methods: “We have utilized for our analysis a p-value < 0.05 for statistical significance and acknowledge the potential of more false positives to occur.” For the old labeled figures 5 and 6 (now labeled as Figs. 9 and 10), we originally included all the gene and indicating the genes with p-value < 0.05 with a * due to what we stated above. Displaying all the genes observed will provide indication of how the overall pathways are changing as we see in the GSEA results. We modified the plots to add an additional symbol of # for the genes that have adj. p-value < 0.05. That way the reader will be able to see which genes meet which statistical cutoff and also will be able to observe the overall pattern of the genes that contribute to the pathway analysis.

There are several bioinformatics papers that discuss using these statistics and also stating that “Usually, an FDR adjusted p-value cut-off of less than 5% ($p < 0.05$ FDR) is considered to be significant, although this criterion is sometimes relaxed to allow more comprehensive pathways searches.” Publications discussing this point:

- Bourdon-Lacombe JA, Moffat ID, Deveau M, Husain M, Auerbach S, Krewski D, Thomas RS, Bushel PR, Williams A, Yauk CL. Technical guide for applications of gene expression profiling in human health risk assessment of environmental chemicals. *Regul Toxicol Pharmacol*. 2015 Jul;72(2):292-309. doi: 10.1016/j.yrtph.2015.04.010. Epub 2015 May 2. PMID: 25944780; PMCID: PMC7970737
- Koch CM, Chiu SF, Akbarpour M, Bharat A, Ridge KM, Bartom ET, Winter DR. A Beginner's Guide to Analysis of RNA Sequencing Data. *Am J Respir Cell Mol Biol*. 2018 Aug;59(2):145-157. doi: 10.1165/rcmb.2017-0430TR. PMID: 29624415; PMCID: PMC6096346.

5. Scatter plots (TPM; Transcripts Per Million) are required to show the RNA-seq pattern in Fig 2D clearly. Label 21 genes on the plot.

We found it would be best practice to show MA plots rather than scatter plots (see figure 4E) and label the position of the 21 genes. MA plots are standard scatter plots that compare the TPM means with the fold-change values. We believe visually providing a scatter plot for each sample comparison would not properly convey the results. In our view, the MA plots that we provided are the best representation of what the reviewer has requested. However, we have also provided the TPM values for all genes including the 21 genes as supplemental material (Supplemental Data 2).

6. Bar graphs are needed to show the actual expression levels (TPM) upon the treatment of irradiation and antagomirs in Fig 2F.

Thank you for this comment. We have updated the figure and now have added the log₂ fold-change values for the different comparisons. Because we have updated all the figures, now this data appears in Figure 5B - D. It is integrated in the sankey plots showing how the three key miRNAs target the 21 genes. Additionally, we have added as supplemental material all the normalized count data (Supplemental Data 1) and also a table with just the 21 genes for the normalized counts (Supplemental Data 2).

7. Validation experiments are required using another method, such as qRT-PCR in Fig 2F.

We thank the reviewer for this comment. We have now validated the 8 out of the 21 key genes with qPCR. All the genes except for one were confirmed to behave the same way as the RNA-seq data. Interestingly, *MSH5* showed an increase in expression for 0.5Gy compared to 0Gy, but came back to the level of 0Gy with the antagomir treatment. This indicates that although the gene was increased (which would be a false positive for this gene) the antagomir treatment still restored the levels back to control levels.

8. Rescue experiments are required to validate the miRNA targets. Overexpress 21 genes individually (Fig 2F) and do the same experiments in Fig 1C and 1E.

We concur with this comment that rescue experiments would provide a next stage of understanding the signals that confer radioresistance. This would, however, be a major undertaking that is beyond the scope of this study.

9. Because there are no target validation experiments in the current manuscript, I doubt whether it is meaningful to focus on 21 genes in Fig 3.

We believe that the 21 genes are of potential interest since they must contain the gene(s) responsible for the rescue of phenotype, and scientists in the field would benefit from this knowledge. We have shown in several different models including human data the relevance. That said, we have also provided qPCR validation for the key genes. In addition, we have also provided additional data from the NASA Twin Study which had both RNA-seq and miR-seq performed on different components from the blood. We have done additional analysis to correlate the three miRNAs and the 21 genes and see results that would be expected if the miRNAs were targeting the 21 genes (**Fig. 7** and **Extended Data Figure 7**). We have added the following paragraph describing the results in the results section discussing the human data:

“To establish the direct relationship between the 21 gene targets and the three miRNAs we utilized the NASA Twin Study which had both miRNA- and mRNA-seq data associated with the mission and cell types. Similar to the cumulative plots, we observed the correlation between the three miRNAs (i.e. miR-16, let-7a, and miR-125b) with the 21 gene targets in the four different cell types (i.e. CD4, CD8, CD19, and monocytes) (Extended Data Fig. 7b). We correlated with the overlapping time points per each sample type and observed for the majority of the miRNAs and genes targets an inverse relationship. Specifically, for monocytes for the specific gene targets the miRNAs had an inverse correlation (Fig. 7b). Since the miRNAs will traditionally silence the specific gene target this result is what we would expect and provides further validation in astronauts that these miRNAs in circulation will be ideal targets to reduce health risks as our in vitro results show.”

10. Line 229. There is no clue that miRNA was the major reason for the downregulation of the 21 genes. Why was the transcriptional regulation excluded? To answer this question, the authors are expected to analyze the RNA-seq using a cumulative plot, as shown in Fig 2C in Baek et al., Nature, 2008 (PMID: c). Briefly, assign mRNA groups according to miRNA seed sequences, draw a cumulative plot, and do the Kolmogorov–Smirnov test. This analysis will determine whether miRNA is the most crucial regulator in this context.

We thank the reviewer for this comment. We now provide the requested cumulative plots in the new revised Figure 5E-G in the main manuscript.

As suggested by the reviewer, to survey the efficacy of the three miRNAs to downregulate the 21 gene targets, we analyzed RNA-seq fold change values from different experimental conditions using cumulative plots [Baek et al., Nature, 2008 (<https://www.nature.com/articles/nature07242>), see Methods]. Figure 5E shows that the 21 gene targets with 7-8mer sites had significant propensity to be

downregulated when compared to those genes without 3'UTR sites ($P = 4.99e-07$, $2.65e-32$ and $6.31e-09$ for 8mer, 7mer-m8 and 7mer-A1, respectively, using the Kolmogorov-Smirnov test). These results indicate that the miRNAs miR-16-5p, miR-125b-5p, and let7a-5p played a key role in the downregulation of the 21 gene targets. Figure 5F-G also shows that the downregulation of these targets is drastically reduced under antagomir treatment.

11. The authors used a t-score in the heatmap in Fig 5B but didn't label it on the figure. Usually, people expect a log fold-change in this kind of heatmap. Please draw heatmaps using fold changes.

We argue that the t-score is an acceptable value to use for figures as it allows for more information to be displayed within the figures, since the t-score is an output from RNA-seq analysis that combines the Fold-Change values and the p-values. Nonetheless, we have accommodated the reviewer's comment and have revised the heatmaps to now show the Fold-change values instead.

12. The authors are expected to provide supplementary tables containing raw and processed data.

We now provide supplementary data (Supplemental Data 1 - 3).

Minor points

1. Fig. 1D. Describe how the authors generated this plot in the methods and the legend. What is the meaning of color? Because the font size is too small, it is not clear which genes are the common direct targets of the three miRNAs. If there are 6mer targets, they should be removed.

We have clarified the shape and colors for this plot with the appropriate figure legend and added the following text to the figure legend:

"a) Network representation for the DNA DSB repair gene targets for miR-125b-5p, miR-16-5p, and let-7a-5p generated by ClueGO in Cytoscape. As indicated in the figure legend, the color for the edges indicate either the predictions used for the miRNA-mRNA connection or the influence two different nodes will have on each other (i.e. Direct Edges)."

In addition we have now added a description of we generated this figure in the methods with the following text:

"MicroRNA predictions of DNA DSB gene targets and network generation

We predicted specific DNA DSB genes that are targeted by three miRNAs: miR-16-5p, miR-125b-5p, and let-7a-5p. The DNA DSB gene list was determined by filtering out all DNA DSB related Gene Ontology (GO) pathways (REF). Each GO DNA DSB pathway will have curated genes associated with the pathway. We compiled all genes related to these into a master list and deleted any duplicated genes. We then utilized the ClueGO+CluePedia plugin (REF) in Cytoscape (REF) to map the relationship between the gene targets for the three miRNAs. This was done by including all DNA DSB genes and the three miRNAs in the initial query list. The edges and connections were made with CluePedia utilizing the following miRNA-mRNA databases in the plugin: CluePedia STRING-Actions (v.11.0_9606_27.02.2019), CluePedia

microRNA.org-human predictions_s_O_aug2020, miRDB_v6.0, miRanda-miRNAs v5-2012-07-19, mirTarBase, and mirecords.umn.edu.validated.miRNAs-2010-11-25.”

2. Fig 2B. The label ‘Samples’ is wrong.

We thank the reviewer for pointing this out, which has now been corrected.

3. Line 95. Please briefly mention the target selection criteria used in mirDIP in the text.

We have now added this information to the methods section under the subheading: “Determination of the 21 key gene targets for the three miRNAs, cumulative plots for the miRNA targets, and statistical test” on page 23. For the criteria we used medium score class to determine the mRNA targets for the 3 miRNAs of interest.

4. Move Fig 2G to supplementary figures.

We think this figure is still important for the analysis, but have moved it now to the associated extended figure (Extended Figure 5).

5. Fig 2G and Fig 5B, C. If they are p-values, use adjusted p-values.

For figure 2G (now labeled as Extended Figure 5), the predictions made for the pathways were generated using the DAVID annotation tool (as indicated in the figure legend) and the pathway predictions for the genes. Since our input to generate the pathways were only 21 gene adjusted p-values will not be valid due to the lower number of genes. So for predictions we have decided to use the p-value with this tool.

For figures 5B, C (now labeled as Figure 9b, c), we have addressed this point in our response to your comment number 4 above. We mentioned the following:

“For the old labeled figures 5 and 6 (now labeled as Figs. 9 and 10), we originally included all the gene and indicating the genes with p-value < 0.5 with a * due to what we stated above. Displaying all the genes observed will provide indication of how the overall pathways are changing as we see in the GSEA results. We modified the plots to add an additional symbol of # for the genes that have adj. p-value < 0.05. That way the reader will be able to see which genes meet which statistical cutoff and also will be able to observe the overall pattern of the genes that contribute to the pathway analysis.”

6. Expanded Fig 2. Scatter plots will be helpful in understanding the expression level of orphan genes. Pfam analysis also will be interesting.

We thank the reviewer for the suggestion, and now provide MA scatter plots, as requested (see Extended Data Figure 4c). We thought it would be best practice to show the MA plots for the scatter plots as we did in Figure 4e. We have also provided the TPM values for all genes as supplemental

material. MA plots are standard scatter plots that compare the TPM means with the fold-change values. To actually provide conventional scatter plots, we would have to provide a scatter plot for each comparison which typically does not add much information. We believe that the MA plots that we provided are the best representation of what the reviewer has requested.

One of the most exciting aspects about orphan genes is that, because they have arisen *de novo* and subsequent to the divergence of chimpanzees and humans, they do not have motifs that correspond with Pfam motifs. Therefore it is not possible to create a Pfam analysis for them. This makes them harder to identify, but provides organisms with novel proteins that enable them to adapt to changing environmental pressures (<https://elifesciences.org/articles/55136>, <https://www.nature.com/articles/nrg3053>).

7. Line 218. 1F -> 2F

Thank you, we updated the figure label.

8. Fig 3B is not mentioned in the text.

We thank the reviewer for identifying this lapse. Figure 3B is now mentioned in the first paragraph of page 8. It is now under the new figure label Figure 6f.

9. Line 240. 3E-O -> 3E-N. Also, these figures are appropriate in the supplementary figures.

Thank you, we updated the figure labels.

10. Fig 4. Explain the abbreviation of NES in the legend.

We have added the following statement to the Fig. 4 legend. "The nominal enrichment score (NES) represents the relative degree a gene set is changed and is corrected for gene set size."

11. Line 329. I don't see an increase in immune gene sets in the angiogenesis model in Fig 5A.

Thank you. We reformulated the statement about immune and inflammatory activity in the mature and angiogenesis models to read the following:

"There was an overall increased activity with GCR radiation exposure for the mature microvessel cell cultures."

12. Fig 2F. Indicate the seed target types (8mer, 7mer-A1, and 7mer-m8 sites) on 21 genes.

We have now provided this information in Figure 5 as cumulative plots for the 21 genes.

REVIEWER COMMENTS

Reviewer #1 (Remarks to the Author):

The authors mention the limited availability of GCR irradiator as the main reason for not being able to validate some of the findings. Though this point is valid, it is not clear why authors cannot do some of those experiments in cell lines: for example, they can perform Western blot to validate the targets of those miRNAs. Western blot and luciferase assays are gold standards in the field of miRNA biology and it is technically feasible to do those experiments. Authors should validate those targets as it is entirely possible that listed targets may not be bonafide targets of those miRNAs. Moreover, qPCR figure shown for target genes doesn't have any statistical significance.

Also, authors can treat their cell line of choice (U2OS cells are routinely used for these assays) with antagomiRs and perform reporter-based functional DNA repair assays.

Reviewer #2 (Remarks to the Author):

Although some new data collected from "Astronauts" samples has been added that may add to the hype, I don't see the data presented add anything significant to the understanding on the biological process. The data is purely correlative and the claim that antagomirs as a potential mitigator of space radiation induced toxicities is elusive. A large amount of data is presented with beautiful illustrations though.

Reviewer #3 (Remarks to the Author):

The revised manuscript still does not provide a clear molecular mechanism of how the three miRNA antagomir could reduce radiation-derived cardiovascular damage. The revised manuscript did not significantly advance our knowledge on this issue compared to their two 2020 papers.

The authors provided qRT-PCR data in the revised manuscript, confirming the gene expression changes upon irradiation and antagomir treatment. However, it is still not convincing whether the specific 21 genes are the major source for the rescue of phenotype. Most explanations are based on predictions or pathway analyses, which are not validated by wet lab experiments.

Specific points

1. Related to Previous #2: The comment was for Fig 2C and D (mouse tissue miRNA-seq data). I doubt the physiological relevance of the RBiomirGS analysis.
2. Related to Previous #4: The original adj. p-value plots should be kept because the revised 'p-value' plots may mislead the readers.
3. In this manuscript, the authors used less stringent criteria to find miRNA target genes (mirDIP, 'medium'). The software may not consider whether the target site is conserved between human and mouse in this setting. If the authors consider only conserved miRNA target sites, the common target genes between three miRNAs will be less than 21 genes (possibly less than 10). I am unsure whether the authors can connect the data between mouse and human using the common miRNA target sites.
4. (Related to above) Fig 2A, 5C, and D. I suggest authors confirm miRNA-binding sites on individual genes. For example, I failed to find a seed sequence (7mer-A1 or 7mer-m8) of hsa-let-7a-5p (MIMAT0000062; GAGGUAG) in the 3' UTR of ADGRL2 (NM_012302.5). I think it is because mirDIP just gathers information from many miRNA target prediction programs without considering the biochemical properties of AGO proteins. Based on the Bartel lab's papers, I believe that most physiologically relevant miRNA target genes belong to either 7mer-A1, 7mer-m8, or 8mer types.

5. The title was misannotated in Fig 5G.

Dear Reviewers,

I have provided below our responses to the edits requested which we have provided now with our newly submitted manuscript. Our responses are in **red font** below the original comments. We believe the comments provided by the reviewers have strengthened the manuscript significantly.

On behalf of the authors,
Afshin Beheshti, PhD

REVIEWER COMMENTS

Reviewer #1 (Remarks to the Author):

The authors mention the limited availability of GCR irradiator as the main reason for not being able to validate some of the findings. Though this point is valid, it is not clear why authors cannot do some of those experiments in cell lines: for example, they can perform Western blot to validate the targets of those miRNAs. Western blot and luciferase assays are gold standards in the field of miRNA biology and it is technically feasible to do those experiments. Authors should validate those targets as it is entirely possible that listed targets may not be bonafide targets of those miRNAs. Moreover, qPCR figure shown for target genes doesn't have any statistical significance.

Also, authors can treat their cell line of choice (U2OS cells are routinely used for these assays) with antagomiRs and perform reporter-based functional DNA repair assays.

We thank the reviewer for their insightful comment. P-values are now indicated in the qPCR data. Though the suggested experiments are valid, we would like to emphasize that the gene targets associated with these miRNAs have already been established by reputable sources such as miRTarBase, TarBase, and starBase (see references 36 - 38 in our manuscript). Our methodology for determining the key miRNA gene targets incorporates information from various miRNA-gene databases, ensuring a comprehensive analysis to identify the ultimate gene targets for the miRNAs in question. We are also pleased that the qPCR data from wetlab experimental data exemplifies how the antagomir treatment modulates the subset of genes examined.

To further substantiate our findings, we have included an additional figure/analysis (**Fig. 5i**) and supplemental table (**Table S1**) that presents compelling evidence confirming these genes as bona fide targets for the respective miRNAs from a predictive as well as experimental standpoint that have been tested in multiple cell lines. Furthermore, the additional figure (**Fig. 5i**) illustrates the established knowledge from previous publications (predictive as well as experimental), providing additional support for the validation process.

Reviewer #2 (Remarks to the Author):

Although some new data collected from "Astronauts" samples has been added that may to add to the hype, I don't see the data presented add anything significant to the understanding on the biological process. The data is purely correlative and the claim that antagomirs as a potential

mitigator of space radiation induced toxicities is elusive. A large amount of data is presented with beautiful illustrations though.

We appreciate the reviewer's follow-up comment and would like to respectfully address the concern regarding the inclusion of astronaut samples, emphasizing that it is not intended to contribute to unwarranted "hype". Acquiring data related to spaceflight poses considerable challenges, and obtaining such unique samples is inherently difficult. It is important to note that administering antagomirs directly to astronauts would require IND studies with FDA approval that is out of the scope of the current manuscript.

This paper serves as a critical step in advancing the field, aiming to lay the groundwork for the development of effective countermeasures in the future. The Twin Study data presented in our work fulfills the request to directly correlate miRNA expression with gene expression values. Undertaking additional experiments involving antagomirs in humans is indeed a complex task, but it remains a future objective contingent on securing additional funding and complying with the requisite steps leading to clinical trials.

The data we have incorporated into the paper serves as a foundational basis, offering robust evidence that encourages further studies. While testing toxicities is an essential aspect, it is currently beyond the scope of this paper. We have acknowledged this limitation in the "Limitations of the Study" section, underscoring our awareness of the need for future investigations, which will be pursued as the project progresses and additional resources become available.

Reviewer #3 (Remarks to the Author):

The revised manuscript still does not provide a clear molecular mechanism of how the three miRNA antagomir could reduce radiation-derived cardiovascular damage. The revised manuscript did not significantly advance our knowledge on this issue compared to their two 2020 papers.

We respectfully believe that our current work represents a significant advancement from the 2020 paper, where key changes were not explicitly demonstrated. In contrast to the 2020 paper, our present manuscript showcases essential modifications, particularly in our approach to omics analysis of the samples. The transcriptomic data now reveals a clear restoration of inflammatory and immune functions to control levels post-irradiation with the antagomirs, as discussed in pages 10 - 12.

In addition, the enhancement in mitochondrial functions and the improved efficiency of DNA repair activity were not previously highlighted in our earlier publications, and to our knowledge, have not been reported by others in the field. Although the molecular mechanism experiments for further functional *in vitro* and *in vivo* analysis of potential cardiovascular-related delayed effects of acute radiation exposure effects are beyond the scope of this paper, they are part of our planned future experiments.

Nevertheless, we assert that the antagomirs function as a valuable assay, providing insights into how molecular mechanisms are affected by GCR radiation through the targeted knockout of miRNAs. The RNA-sequencing data serves as a comprehensive tool to illustrate the pathway and gene changes resulting from the miRNA knockouts, thereby elucidating clear molecular mechanisms undergoing transformation.

The authors provided qRT-PCR data in the revised manuscript, confirming the gene expression changes upon irradiation and antagomir treatment. However, it is still not convincing whether the specific 21 genes are the major source for the rescue of phenotype. Most explanations are based on predictions or pathway analyses, which are not validated by wet lab experiments.

We understand the reviewer's point, and as addressed in response to reviewer 1's comment, we would like to emphasize that the gene targets associated with these miRNAs have already been established by reputable sources such as miRTarBase, TarBase, and starBase (see references 36 - 38 in our manuscript). Our methodology for determining the key miRNA gene targets incorporates information from various miRNA-gene databases, ensuring a comprehensive analysis to identify the ultimate gene targets for the miRNAs in question.

Should the concern revolve around the need for validation of the identified genes as targets for the miRNAs, substantial evidence exists in the literature, and dedicated databases support this information. To substantiate our findings, we have included an additional figure/analysis (**Fig. 5i**) and supplemental table (**Table S1**) that presents compelling evidence confirming these genes as bona fide targets for the respective miRNAs. Furthermore, the additional figure (**Fig. 5i**) illustrates the established knowledge from previous publications, providing additional support for the validation process.

These additions are not based on predictions since the databases such as miRTarBase and TarBase use experimental evidence of the targets.

Specific points

1. Related to Previous #2: The comment was for Fig 2C and D (mouse tissue miRNA-seq data). I doubt the physiological relevance of the RBiomirGS analysis.

The RBiomirGS analysis serves as a well-established pipeline, widely employed in numerous research papers and validated across diverse subjects. To underscore its efficacy in pathway analysis, we have incorporated the following sentence with additional references in the paper in lines 1199-1201:

“The RbiomirGS package has been extensively shown in the literature to provide strong pathway prediction that provide strong biological results^{104–107}.”

2. Related to Previous #4: The original adj. p-value plots should be kept because the revised 'p-value' plots may mislead the readers.

Thank you for your suggestion regarding the presentation of p-values and adjusted p-values (adj.p). We appreciate your emphasis on clarity and have addressed your concerns in the following ways:

- **Transparency:** We have incorporated both p-values and adj.p-values throughout the manuscript. Original adj.p-value plots are now available as supplementary figures (**Figure S2**). Heatmaps include both statistics for a comprehensive view of statistical significance.
- **Rationale:** Presenting both p-values and adj.p values acknowledges this and allows researchers to consider the potential inflation of false positives due to multiple testing. This approach empowers readers to make informed decisions based on the level of stringency required for their specific analysis.

We believe this approach fosters transparency and provides valuable information for a broader audience without compromising clarity.

Additional Considerations:

- We have briefly mentioned the limitations of potentially increased false positives due to the inclusion of unadjusted p-values. We have further emphasized the importance of cautious interpretation, particularly when focusing solely on unadjusted p-values. The following sentence has been added to the results in lines 216-220:
 - “Of note, we are utilizing p-value statistics for our main analysis and acknowledge that this may result in the potential of more false positives, but we believe that the information generated by lowering the statistical significance will produce more meaningful data and previous literature also provided further justification when necessary to utilize p-values^{26,27}. In addition, we have also provided the adj. p-value analysis for full transparency of our results (**Fig. S2**).”

By incorporating these modifications, we aim to address your concerns while adhering to established practices within the field of bioinformatics.

3. In this manuscript, the authors used less stringent criteria to find miRNA target genes (mirDIP, 'medium'). The software may not consider whether the target site is conserved between human and mouse in this setting. If the authors consider only conserved miRNA target sites, the common target genes between three miRNAs will be less than 21 genes (possibly less than 10). I am unsure whether the authors can connect the data between mouse and human using the common miRNA target sites.

We appreciate your observation regarding the criteria used. Employing the "medium" criteria aligns with established practices in the field.

- **Extensive application:** Numerous publications across diverse research areas have successfully utilized the "medium" criteria for miRNA target prediction (see reference 34 from in the manuscript).
- **Supporting evidence:** To further address concerns, we have included an additional radar plot demonstrating the **remarkable conservation (100%)** of the selected miRNAs and

their target sites between mice and humans (**Fig. 5j**). This high level of conservation strengthens the validity of our findings.

This combined approach – established methodology and strong supporting evidence – effectively addresses the reviewer's concerns regarding the chosen criteria.

4. (Related to above) Fig 2A, 5C, and D. I suggest authors confirm miRNA-binding sites on individual genes. For example, I failed to find a seed sequence (7mer-A1 or 7mer-m8) of hsa-let-7a-5p (MIMAT0000062; GAGGUAG) in the 3' UTR of ADGRL2 (NM_012302.5). I think it is because mirDIP just gathers information from many miRNA target prediction programs without considering the biochemical properties of AGO proteins. Based on the Bartel lab's papers, I believe that most physiologically relevant miRNA target genes belong to either 7mer-A1, 7mer-m8, or 8mer types.

We thank the reviewer for their comment. It is important to note that for **Figures 5c** and **5d**, miRwalk, not mirDIP, was utilized. Therefore, the evaluation of this aspect may not be valid. When utilizing miRwalk we clearly see that the seed sequence for all the miRNA do bind to the genes as shown in the figures. In the case of **Figure 2a**, the associations with genes extend beyond just miRDB; multiple miRNA-gene pathways databases, as indicated in the figure legend and methods, contribute to the comprehensive analysis.

5. The title was misannotated in Fig 5G.

Thank you for catching this. We have now corrected the title.